



# The Greenland Ice Sheet Large Ensemble (GrISLENS): Simulating the future of Greenland under climate variability

Vincent Verjans[1,2*], Alexander A. Robel[2*], Lizz Ultee[3], Helene Seroussi[4], Andrew F. Thompson[5], Lars Ackermann[6], Youngmin Choi[7], and Uta Krebs-Kanzow[6]

[1]Barcelona Supercomputing Center, Barcelona, Spain
[2]School of Earth and Atmospheric Sciences, Georgia Institute of Technology, Atlanta, GA, USA
[3]NASA Goddard Space Flight Center, Greenbelt, MD, USA
[4]Thayer School of Engineering, Dartmouth College, Hanover, NH, USA
[5]Environmental Science & Engineering, California Institute of Technology, Pasadena, CA, USA
[6]Alfred Wegener Institute Helmholtz Centre for Polar and Marine Research, Bremerhaven, Germany
[7]Earth System Science Interdisciplinary Center, University of Maryland, College Park, MD, USA
[*]Both authors equally contributed to this study

**Correspondence:** Alexander A. Robel (robel@eas.gatech.edu)

**Abstract.** The Greenland Ice Sheet has lost ice at an increasing pace over recent decades, driven by a combination of human-caused climate change and internal variability of the climate system. In projections of future ice sheet evolution, internal variability of climate results in uncertainty that cannot be reduced through model improvements, due to the intrinsically chaotic nature of the climate system. This study describes the Greenland Ice Sheet Large Ensemble (GrISLENS), the first large ensem-

ble study of ice sheet evolution under climate variability which resolves individual outlet glaciers as well as climate variability calibrated to observations. GrISLENS combines multiple advanced modeling methods, including a stochastic ice sheet model, a coupled atmosphere-ocean model, dynamical surface mass balance downscaling, and statistical techniques for constraining stochastic parameterizations of climate forcing. We quantify the role of internal climate variability in 185-year projections of the Greenland Ice Sheet under both a high-emission scenario and pre-2000 climate conditions. We find that spread between

ensemble members due to internal climate variability represents a substantial fraction of the mean ice sheet change in the first 20-30 years of simulations, which may be important for coastal planning efforts on decadal time scales. This spread between ensemble members reduces to a small fraction of the total ice sheet change past 2050. At the ice-sheet scale, uncertainty in ice loss is dominated by the response to surface mass balance variability, while the response ocean variability is relatively small, though its influence is more important within individual catchments. The GrISLENS ensemble spread is relatively small

compared to previous studies estimating uncertainty from climate variability in coarse models, which indicates that resolving small scale features in climate forcing and ice sheet dynamics substantially affects the quantification of internal variability in ice sheet mass change. On longer time scales, human emissions of greenhouse gases and structural and parametric uncertainties in climate and ice sheet models are larger contributors to projection uncertainties. Through our analysis, we identify the need for more robust initialization methods, as well as multi-centennial large-ensemble simulations that sample internal variability

to the Antarctic Ice Sheet.





## 1 Introduction

Mass loss from the Greenland Ice Sheet (GrIS) has contributed ∼16 % of total global sea-level rise since 1992 (IPCC, 2021), driven by increasing surface melt (Fettweis et al., 2016) and accelerated ice discharge into the ocean (King et al., 2020). GrIS mass loss is also rapidly accelerating: the mass loss from 2012 to 2020 was more than five times the mass loss from 1992 to 2000 (Otosaka et al., 2023). Over the 21st century, the projected contribution of the GrIS to sea-level rise is sensitive to both the greenhouse gas emission scenario and internal variability of the climate system (Tsai et al., 2017). While quantifying the ice sheet sensitivity to greenhouse gas emissions has been the focus of substantial community efforts (Nowicki et al., 2013; Goelzer et al., 2020), less attention has been paid to quantifying the range of possible future ice sheet evolution due to internal variability of the climate system alone.

Projections of future GrIS mass balance are generally determined from ice sheet model simulations, with prescribed climatic forcing (e.g., Goelzer et al., 2020). A number of factors contribute to uncertainties in ice sheet model projections (Aschwanden et al., 2021). First, structural model uncertainty stems from our incomplete understanding of processes regulating ice sheet dynamics, such as iceberg calving (Amaral et al., 2020), subglacial hydrology (Kazmierczak et al., 2022), basal sliding (Choi et al., 2022), and ice mélange (Joughin et al., 2020). Second, some processes are not fully resolved by the typical spatial resolution and time steps in ice sheet models. This can be due to insufficient knowledge of fine-scale mechanisms or computational limitations. Such processes are parameterized using physical mechanisms and observational constraints, but the high number of unknown parameters in such calibration procedures implies that numerical parameter values remain uncertain (Wernecke et al., 2020; Berends et al., 2023). Third, uncertainty in initial conditions impacts simulations over long periods, and therefore projections (Adalgeirsdottir et al., 2014; Yang et al., 2022). Imperfect knowledge of the current and past dynamical states of the GrIS has a long-lasting influence on ice sheet evolution as well as observational errors in ice sheet geometry and other observational fields (Seroussi et al., 2011). To evaluate the total uncertainty arising from these sources, ice sheet model intercomparisons are performed, in which each participating model may include different physical processes, initialization methods, parameter schemes, and numerical schemes (Goelzer et al., 2020). The results of such intercomparisons (the most recent being ISMIP6, Goelzer et al., 2020) have been used as the primary source for sea level projections of the Intergovernmental Panel on Climate Change (IPCC, 2021) and related efforts (Edwards et al., 2021).

Projections of ice sheet mass balance are also influenced by the inherent uncertainty associated with future climate variability (Mikkelsen et al., 2018; Robel et al., 2019). For any given scenario of anthropogenic forcing, there remains an "irreducible" (also called "aleatoric") uncertainty associated with the ice sheet response to internal climate variability, due to the fundamental unpredictability of the climate system (Lorenz, 1969). It is common practice in the climate modeling community to quantify internal variability through large ensembles, sampling different initial conditions as well as multiple realizations of each climate model (Mankin et al., 2020; Deser et al., 2020). However, current ice sheet model intercomparisons are not designed to quantify this aleatoric uncertainty, as they typically force ice sheet models with a single realization from each combination of climate models and emissions scenario selected (e.g., Goelzer et al., 2020; Li et al., 2023). Yet, idealized studies have demonstrated the sensitivity of glacier evolution to processes that have comparable time scales inherent to internal climate variability (Roe and



Baker, 2016; Mikkelsen et al., 2018; Robel et al., 2018; Christian et al., 2020; Verjans et al., 2022). For example, processes, such as atmospheric blocking, that are poorly represented in climate models (Hanna et al., 2018) might have an important effect on Greenland Ice Sheet projections (Beckmann and Winkelmann, 2023). Despite this known strong sensitivity of the GrIS to climate forcing, sensitivity to internal climate variability and associated irreducible uncertainties remain poorly quantified (Aschwanden et al., 2021).

To understand the importance of internal climate variability for ice sheet mass balance over the next 200 years, large ensemble experiments are a viable approach. In recent years, Global Climate Model (GCM) large ensembles have been produced and made available as community data sets (e.g., Kay et al., 2015; Rodgers et al., 2021). This has enabled the investigation of many questions, such as possible future changes in modes of climate variability (Zheng et al., 2018), sensitivity of the overall climate variability spectrum to forcing (Rodgers et al., 2021), attribution of extreme weather events to anthropogenic forcing

(Diffenbaugh et al., 2017), and evaluation of the range of possible socioeconomic impacts of climate change (Schwarzwald and Lenssen, 2022). Internal climate variability has also been extensively explored through observational studies (Mitchell, 1976; McKinnon et al., 2017). This inherent feature of the climate system therefore deserves attention, including its impacts on evolution of the GrIS.

Tsai et al. (2017) attempted the only prior study on the role of internal climate variability on the evolution of the GrIS. Their
approach used 40 and 50 members of two different coarse GCM large ensembles as direct climate forcing for Greenland ice sheet model simulations over the 21st century. They found that, under a high-emission scenario, the spread in sea-level rise contribution by 2100 between ensemble members is ∼17 % of their ensemble-mean simulated ice sheet mass loss. However, this method faces two limitations. First, the coarse resolution of GCM outputs (∼100 km grid scale, monthly time steps) implies limited representation of processes with strong spatial gradients. Second, GCM outputs do not correspond directly
to inputs needed for ice sheet models, and thus some assumptions have to be made for such conversions, such as simplified empirical relationships between atmospheric temperature and ice sheet surface mass balance. Furthermore, Tsai et al. (2017) ran their ice sheet model at a coarse resolution over the GrIS (20 km), therefore potentially not resolving individual outlet glaciers, almost all of which are less than 20 km wide in Greenland. Use of such a coarse resolution has been shown to induce substantial quantitative impacts on ice sheet model results (Greve and Herzfeld, 2013). Still, the results of Tsai et al. (2017)
have demonstrated that uncertainty associated with internal climate variability may amount to a non-negligible fraction of future GrIS change and persist on decadal to centennial time scales.

In this study, we describe the Greenland Ice Sheet Large Ensemble (GrISLENS), a modeling experiment estimating the GrIS sensitivity to internal climate variability down to sub-kilometer scales. We apply a novel stochastic modeling approach to address this question. Specifically, we calibrate spatio-temporal stochastic models to downscale GCM outputs, and use them
subsequently to force ice sheet model simulations spanning 2018 to 2203. Our results quantify model spread in GrIS mass change, ice thickness change, and glacier retreat, all associated with internal climate variability. We compare this spread with ensemble mean change under different mean forcings, as well as with differences in projected ice sheet mass between different ice sheet and climate models from prior intercomparison studies. Finally, we make our model output openly available to enable



future investigations into the role of internal climate variability in driving ice sheet change, similar to the growing use of large
ensemble climate model outputs in climate sciences.

## 2   Methods

The ensemble of simulations presented in this study is the culmination of prior work to develop methods which can stochasti-
cally generate realistic climate forcing for ice sheet models down to sub-kilometer spatial scales. At the base of these methods
is output from a climate model spanning both the historical period over which the ice sheet model is initialized and a pe-
riod of several centuries into the future over which the evolution of the GrIS is expected to unfold. Atmospheric forcing is
then converted and downscaled to surface mass balance (SMB) using a high-resolution energy-balance model (Krebs-Kanzow
et al., 2020), and used to train a stochastic SMB parameterization for every glacier catchment in Greenland using the method
described by Ultee et al. (2024). Ocean forcing is calibrated to available observations of ocean properties around Greenland,
downscaled to fjords, and then used to train stochastic parameterizations of ocean thermal forcing for every marine-terminating
glacier catchment in Greenland using the method described by Verjans et al. (2023). The stochastic parameters of these meth-
ods are then input directly into the Stochastic Ice-Sheet and Sea-Level System Model (StISSM, Verjans et al., 2022), which
internally generates realistic and correlated climate variability to force the ice sheet model equations. While the details of these
methods can be found in the studies cited above, additional methodological developments needed for this study are described
below.

### 105   2.1   Climatic forcing

#### 2.1.1   Reference Climate Model Simulation

Atmospheric and ocean forcing for our Greenland simulations is based on a 1850-2203 simulation of the Alfred Wegener In-
stitute Earth System Model (AWI-ESM, Ackermann et al., 2020). The ocean model implemented in AWI-ESM is FESOM1.4
(Wang et al., 2014). It employs an unstructured grid, with a resolution of ~20 km around Greenland. The atmosphere is repre-
sented with the ECHAM6 model (Stevens et al., 2013), using a horizontal resolution of $1.85°$ (50-100 km across Greenland).
After the historical period 1850-2005, the simulation that we use follows the high-emission scenario RCP8.5 (Riahi et al.,
2007) until 2100, and keeps the atmospheric $CO_2$-equivalent forcing fixed at the 2100 level for the 2101-2203 period. We note
that, in recent GCM intercomparisons, AWI-ESM shows a behavior close to the multi-model average in terms of key climatic
features, such as equilibrium climate sensitivity and transient climate response (Semmler et al., 2020).

#### 115   2.1.2   Atmospheric forcing

AWI-ESM atmospheric fields over Greenland are downscaled at 5 km horizontal resolution, using the diurnal Energy Balance
Model (dEBM, Krebs-Kanzow et al., 2020). In a recent intercomparison of SMB from different regional climate models
(Fettweis et al., 2020), dEBM GrIS-averaged SMB was shown to be less than 20% of the ensemble standard deviation from





the ensemble mean SMB and to display temporal dynamics within the uncertainty range of gravimetry observations. We use

the SMB and runoff output fields from dEBM and we process these variables at the catchment-level. Specifically, we use the 253 catchment delimitations over the contiguous Greenland Ice Sheet of Mouginot et al. (2017). We take the average SMB and the total runoff over each catchment, and aggregate the monthly values at an annual resolution. As such, each catchment has a single 1850-2203 annual time series for both SMB and runoff, which can serve to calibrate stochastic time series in the models, as detailed in Section 2.2 (see also Ultee et al., 2024). In this study, we consider SMB as an annual variable, as sub-annual

variability in SMB is unlikely to play a strong role on ice sheet dynamics over decadal to centennial time scales (Robel et al., 2019; Christian et al., 2020). However, we do account for monthly variability in runoff by computing the average fraction of the annual runoff occurring at each month for each individual catchment, since monthly runoff affects melt at the ocean interface (see Section 2.2.3). Within-catchment spatial variability of SMB is captured through estimation of the slope of SMB-elevation profiles, as described in Ultee et al. (2024). Further detail and examples for this specific study can be found in Appendix B.

### 2.1.3   Ocean forcing

Ocean forcing for the ice sheet model is derived from variations in ocean temperature and salinity simulated by AWI-ESM. In our ice sheet modeling framework, the ocean melt rate parameterizations uses thermal forcing ($TF$), the water temperature above freezing point. $TF$ is integrated through depth to calculate parameterized melt rates at glacier fronts. As waters around Greenland are stratified, generally with cold and fresh Arctic water above warmer and saltier Atlantic waters (Straneo et al.,

2012), the depth range over which $TF$ is integrated influences strongly the melt rates calculated. We find the effective depth of each Greenland marine-terminating glacier, which is defined as the deepest bathymetry connected to the open ocean without a barrier (e.g., Morlighem et al., 2019; Slater et al., 2020). We use the BedMachine v4 product for the bathymetry around Greenland (Morlighem et al., 2017). For each marine-terminating glacier, we integrate $TF$ from the surface until its effective depth (which does not vary with ice sheet evolution), and each glacier is thus assigned a specific field of depth-integrated $TF$

time series. For a detailed discussion of this choice of method for calculating $TF$, see Appendix C. Throughout this study, we use the notation $TF$ to refer to this depth-integrated value. We further adjust the AWI-ESM $TF$ following a two-step procedure, as summarized below and detailed at length in Verjans et al. (2023).

First, we bias-correct the AWI-ESM $TF$ based on the EN4 ocean monthly objective analyses (Good et al., 2013). For each AWI-ESM grid point, we perform the bias-correction with the nearest EN4 grid point using the 1950-2006 $TF$ from AWI-

ESM and EN4, which corresponds to the pre-RCP8.5 forcing in AWI-ESM. The period before 1950 is not used for the bias-correction because EN4 is poorly constrained by observations (Verjans et al., 2023). The bias-correction follows a Quantile-Delta-Mapping procedure (Cannon et al., 2015; Verjans et al., 2023). This procedure calibrates both the mean and amplitude of variability of AWI-ESM to EN4, while preserving the relative changes in time of $TF$ as modeled by AWI-ESM.

Second, we downscale the bias-corrected $TF$ from the AWI-ESM grid to fjord mouths. The extrapolation is based on statisti-

cal relations found between shelf waters and fjord mouth conditions in output of a high-resolution ocean model reanalysis, ECCO2-Arctic (Nguyen et al., 2012). These statistical relations are derived for the long-term mean $TF$, as well as for the seasonal and non-seasonal variability. The fjord mouth locations are selected as the closest ECCO2-Arctic grid point to each





Greenland glacier front from Wood et al. (2021), with bathymetry at least as deep as the effective depth. The data set of glacier fronts includes 226 distinct marine-terminating glaciers.

All our climatic forcing procedures are performed at the catchment-scale, but some catchments include no or more than one marine-terminating glaciers. As each catchment is assigned a single set of *TF* statistics, we select a single *TF* series for catchments with more than one marine-terminating glacier. We select the *TF* series of the glacier with deepest effective depth, because these glaciers have generally more impact on the total mass balance of the Greenland Ice Sheet. As such, there are 50 glaciers which are assigned a TF time series of a neighboring glacier with deeper effective depth. For these 50 pairs of time

series the 0.5, 0.75, and 0.95 quantiles of root mean square deviation are 0.7, 1.5, and 2.1 K, respectively. For the 77 catchments without marine-terminating glaciers, *TF* does not need to be prescribed.

## 2.2   Stochastic representation of climatic forcing

### 2.2.1   Calibration of stochastic models

Starting from the AWI-ESM outputs, three climatic variables, with time series for each glacier catchment are post-processed:

*TF*, SMB and runoff. We separate each annual time series into a deterministic and a stochastic component. The former accounts for the mean forcing and trends, e.g., under the RCP8.5 emissions scenario. The latter accounts for the irreducible uncertainty associated with natural climate variability, and is the residual obtained from the original time series by removing the deterministic component. For fitting stochastic time series models, the residual variability should be stationary and homoskedastic, i.e., without trends and with constant variance over time (von Storch and Zwiers, 1999). For our three variables, we find that resid-

ual variability is stationary if we account for deterministic components that are piecewise-linear in time (breakpoints in 2000, 2050, and 2100) and normalize the variance of the residuals (creating a normalized "z-score" series) to account for change in the amplitude of variability over time for each of the three sub-periods. The breakpoints are chosen to capture periods of change in the mean and variability amplitude of climate forcing, such that normalized variability is stationary (see below). We chose 2100 as a breakpoint since emissions are held constant after this point in the RCP8.5 climate model simulation described

above. We find that just one more breakpoint is needed between 2000 and 2100 to capture the change in the mean and variability amplitude of climate forcing, and for expediency choose the midpoint of this period. An example of this is illustrated in Figure 1 for Humboldt glacier (northeast Greenland). The original annual time series of our three variables, shown in blue, are well-represented by piecewise-linear time series, shown in red. The difference between them is the residual variability, shown in green. The latter is well-centered around 0, showing that the secular trends are correctly captured by the piecewise-linear

functions with the specified break points. However, the residual variability time series clearly show increasing variance in time. Once normalized, the resulting z-score time series are stationary, trendless and homoskedastic across time.

To validate our approach for generating standardized residual variability time series, we use the Augmented Dickey-Fuller test (Dickey and Fuller, 1981). We have 253 time series for SMB, 247 for runoff because 6 catchments have zero runoff over the entire simulation period, and 176 for TF because 77 catchments have no marine-terminating glaciers, thus resulting in





676 z-score time series in total. The null hypothesis of non-stationarity in the Augmented Dickey Fuller test is rejected with significance for all the 676 z-score time series (p-values<0.05).

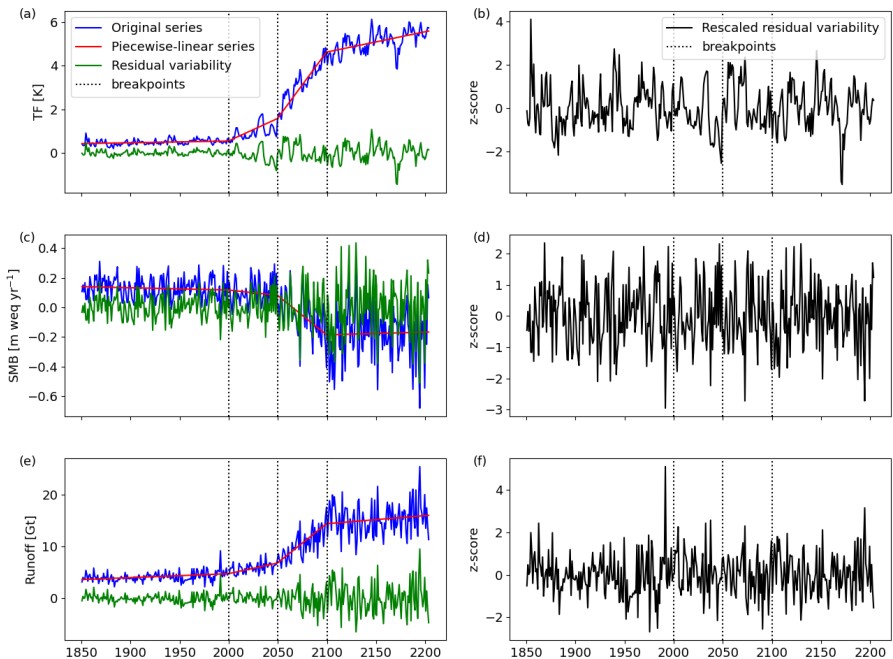

**Figure 1.** Annual climatic forcing time series at the catchment of Humboldt Glacier (see Fig. 7a for location). Original forcing time series are in blue, fitted piecewise-linear functions in red, and residual variability in green. The residual variability is obtained by subtracting the piecewise-linear function from the original series. Breakpoints of the piecewise-linear functions are shown with dotted lines. The residual variability is then standardized to unit variance separately in each sub-period to have unit variance. Standardized residual variability time series are shown in black, in the right column. Bias-corrected and extrapolated *TF* (a), and its rescaled residual variability (b). SMB (c), and its rescaled residual variability (d). Runoff (e), and its rescaled residual variability (f).

We fit stochastic time series models to the annual z-score time series. More specifically, we calibrate an Autoregressive-Moving-Average (ARMA) model to each individual time series. ARMA processes are efficient representations of climatic variables, as they can capture an extensive range of time scales, while using a small number of parameters (Hasselmann, 1976; von Storch and Zwiers, 1999; Wilks, 2011). Thus, by representing the residual variability component of our climatic variables

of interest as an ARMA process, we aim to capture the inter-annual to decadal time scales of climate variability that force the Greenland Ice Sheet climate. Further discussion on the validity of using ARMA models to capture SMB and *TF* variability can be found in Ultee et al. (2024) and Verjans et al. (2023), respectively.

For each time series, we calibrate all possible combinations of ARMA models with autoregressive (AR) orders and moving

average (MA) orders ranging from 0 to 4. We use the well-established Bayesian Information Criterion (BIC, Schwarz, 1978) to select ARMA models best fitting the target time series, with a penalty proportional to the number of parameters used. Figure





A1 shows the selected ARMA combinations for our three climatic variables, and for all the catchments. A large majority of catchments have their SMB and runoff z-score time series best described as an inter-annual white noise process, which corresponds to an ARMA(0,0) process, i.e., without memory of previous years. Only 39 of the 253 SMB and 15 of the 247 runoff z-score series include a lag term (i.e., the $p$ order of ARMA$(p,q)$). In contrast, only 19 of the $TF$ z-score series are best fitted by annual white noise. Most (70%) have an ARMA(1,0) as their best-fitting model, and 6% include a second-order lag term (i.e., ARMA$(2,q)$; Fig. A1).

We account for correlation between all the z-score time series, such that time series stochastically generated with ARMA models reproduce the desired level of inter-dependence between catchments and climatic variables (e.g., Ultee et al., 2024). For this purpose, we calculate the empirical correlation matrix between the residuals of the fitted 676 z-score time series. The residuals are obtained after fitting the optimal ARMA model to a given time series, such that the stochastic component $\epsilon_t$ (see Eq. (A2)) is isolated. However, because the number of entries in the correlation matrix to be estimated ($\frac{1}{2} \times 676 \times 675 = 228\,150$) is large compared to the number of yearly samples (354), we compute a sparse correlation matrix (Hu and Castruccio, 2021) with the commonly-used graphical lasso method (Friedman et al., 2008). For all variables, correlation patterns are strong within East- and West-Greenland, but weaker between East- and West-Greenland. The cross-correlation between $TF$ and the two other climatic variables is very low, while SMB and runoff are anti-correlated, as expected. To generate stochastic annual time series of our three climatic variables, we use different covariance matrices for our different sub-periods separated by the 2000, 2050, and 2100 breakpoints. All covariance matrices share the same correlation structure described above. However, the covariance magnitudes are scaled to match the different amplitudes of variability between the different periods, and this procedure is detailed in the Appendix.

At the end of our processing of the annual climatic time series, we have derived deterministic piecewise-linear functions to capture the long-term mean and trends in $TF$, SMB, and runoff. In addition, we have calibrated all the components of the stochastic time series models (see Eqs. (A2, A3)) to represent the spatiotemporal climatic variability, accounting for inter-variable and inter-catchment covariability. Finally, we have also derived catchment-specific lapse rates to capture the elevation dependence of SMB, which is important to account for within-catchment variability of SMB, and for the SMB-elevation feedback as the ice sheet geometry changes during the long simulation period (Edwards et al., 2014; Ultee et al., 2024).

### 2.2.2 Sub-annual variability

In general we neglect sub-annual variability in SMB because high-frequency variability in SMB exerts a minor influence on decadal and longer time scales ice dynamics compared to other forcings (Robel et al., 2019; Christian et al., 2020; Ultee et al., 2022). However, we account for seasonal variability in both runoff and $TF$, for which short-term variability exerts a stronger control on the evolution of marine terminating glaciers (Felikson et al., 2022; Slater and Straneo, 2022; Ultee et al., 2022)

For $TF$, we follow Verjans et al. (2023) and add a climatological anomaly calculated from AWI-ESM output as the anomaly from the long-term mean for each calendar month and for each marine-terminating catchment. However, we observe that seasonality changes strongly over the period of our simulations. This is particularly true for marine-terminating glaciers in North Greenland, for which the zero-bound on $TF$ due to sea-ice presence progressively vanishes for an increasing number of





months. To incorporate this changing seasonality, we fit piecewise linear functions for each monthly climatological anomaly with breakpoints in 2000, 2050, and 2100.

For runoff, we compute the fraction of the total annual runoff occurring in each month for each catchment averaged over the multi-decadal period between breakpoints (2000-2050, 2050-2100, 2100-2203). As such, the 12 monthly fractions sum to

1 and are catchment-specific. Since annual total runoff is concentrated in just a few months and is otherwise zero, we calculate runoff seasonality to prevent spurious winter runoff. In our ice sheet model simulations, the annual runoff is distributed over any given year according to these fractions.

### 2.2.3   Ice sheet model forcing

For our simulations, stochastic realizations of the three climatic variables are generated within the ice sheet model, the Stochas-

tic Ice-sheet and Sea-level System Model (StISSM, Verjans et al., 2022). StISSM generates noise, computes the corresponding time series (Eq. (A2)), and performs the SMB lapse rate corrections within the evolving ice sheet simulation. Doing these computations within the ice sheet model is preferable to capture feedbacks with ice sheet geometry (on SMB in particular) and correlations between climate forcing variables, and to avoid uncertain conversions from inputs to ice sheet model quantities. Runoff and $TF$ are used as forcings for a parameterization of frontal melt rate ($\dot{m}_{fr}$) at grounded marine-terminating glacier as

described by Rignot et al. (2016):

$$\dot{m}_{fr} = (Ah_w q_{sg}^{\alpha} + B)TF^{\beta}, \tag{1}$$

where $q_{sg}$ is the subglacial water flux [m day$^{-1}$] and $h_w$ is water depth [m]. The calibration parameter values are $A = 3{\times}10^{-4}$, $B = 0.15$, $\alpha = 0.39$, and $\beta = 1.18$ (Rignot et al., 2016). We substitute the catchment-integrated runoff generated by StISSM for $q_{sg}$, thus assuming that all the runoff over a given time step is discharged immediately at the marine front in catchments in

contact with the ocean.

For floating ice, we follow a simplification of the "three-equation" melt parameterization (Holland and Jenkins, 1999; Beckmann and Goosse, 2003):

$$\dot{m}_{fl} = \rho_w c_{pM} \gamma_T F_m TF, \tag{2}$$

where $\rho_w$ is the ocean water density (1023 kg m$^{-3}$), $c_{pM}$ is the ocean mixed layer specific heat capacity (3974 J kg$^{-1}$ K$^{-1}$),

$\gamma_T$ is the thermal exchange velocity ($10^{-4}$ m s$^{-1}$), and $F_m$ is an empirical melt factor set to 0.203 following Favier et al. (2019).

### 2.3   Model calibration and transient initialization

Prior to running stochastic transient simulations until 2203, we initialize the ice sheet model and calibrate model parameters to match the ice sheet state and its transient evolution over the period 2007-2017.





### 2.3.1 Model initialization

We configure the Greenland Ice Sheet initial state with the bed topography, ice thickness, and ice mask from BedMachine v4 (Morlighem et al., 2017), the 2007 ice velocity field from Joughin et al. (2010), the geothermal heat flux from Shapiro and Ritzwoller (2004), and surface temperature from Ettema et al. (2009). To approximate the stress balance equation and enable many long simulations in the ensemble, we use the Shallow Shelf Approximation (Macayeal, 1989). We solve a thermal steady-state model in three dimensions with 10 vertical levels, and compute vertical temperature profiles. The ice rheology field is then calculated following the temperature-dependent parameterization of Cuffey and Paterson (2010), and is subsequently depth-averaged to be used in the two-dimensional model configuration of this study. Basal friction is set by the Budd sliding law (Budd et al., 1979):

$$\boldsymbol{\tau_b} = -C_b^2 \boldsymbol{u_b} N, \tag{3}$$

where $\boldsymbol{\tau_b}$ is the basal stress [Pa], $\boldsymbol{u_b}$ is the basal ice velocity [m yr$^{-1}$], and $C_b^2$ is the basal friction coefficient [m$^{-1}$ yr]. In areas with ice thickness larger than 500 m, we invert for $C_b^2$ based on the observed velocity field. We perform a linear regression of the inverted $C_b^2$ field with respect to bed topography to calculate $C^2$ in regions where ice thickness is <500 m or absent, which allows possible expansion of the ice sheet during the transient simulations.

The domain is meshed with a variable horizontal resolution, ranging from 25 km in the slow-flowing interior of the GrIS, to less than 1 km in the fastest flowing areas at the ice sheet edge. We simulate dynamic calving front migration at 148 marine-terminating glaciers using the level set method, and we apply streamline upwinding for numerical stability (described in Bondzio et al., 2016). Choi et al. (2021) identified these 148 glaciers as having sufficiently well-constrained bathymetry from the data set of Wood et al. (2021). For the other, mostly smaller, 78 glaciers of the data set of Wood et al. (2021), we keep the calving front fixed during the simulations. In the vicinity of the 148 dynamic glacier fronts, we refine the mesh resolution to 800 m. We cannot predict how far glaciers retreat by the end of the simulation period a priori, and thus how far inland the refined mesh should be extended in order to accommodate glaciers' retreat. We therefore use a pragmatic approach by extending the refined regions several 10s or even 100s of kilometers inland for the largest glaciers with most influence on the total GrIS mass balance. For the majority of small marine-terminating glaciers, we limit the refined regions to 10-30 km inland, in order to limit computational expense associated to the very high mesh resolution. A posteriori, we find that in our simulations with strongest warming and thus highest retreat rates, 9 of the 148 marine-terminating glaciers retreat up to their limit of refined region. No retreat beyond that point is simulated for these glaciers, and the simulated ice mass loss estimations from these 9 relatively small glaciers is therefore affected. Despite this limitation, our adoption of limited refined regions is a reasonable compromise because ice discharge from the GrIS is dominated by a small (<20) number of large glaciers (Enderlin et al., 2014). At none of these largest 20 marine-terminating outlet glaciers, does retreat extend beyond the region of refined mesh.





### 2.3.2 Calibration

After the initialization, we perform a short 11-year calibration run over the period 2007-2017. This calibration does not explicitly include stochasticity in the climate forcing, but does include the climate forcing exactly as simulated by AWI-ESM for this time period. For runoff, we use the annual dEBM output over 2007-2017, including the annual cycle. For SMB, we use

SMB-elevation profiles fit from dEBM output (Sect. 2.2.1) to describe the mean forcing over 2007-2017 in each catchment and downscale SMB onto the model mesh. For *TF*, we use the extrapolated and bias-corrected output of AWI-ESM (see Sect. 2.1.3) during this time period.

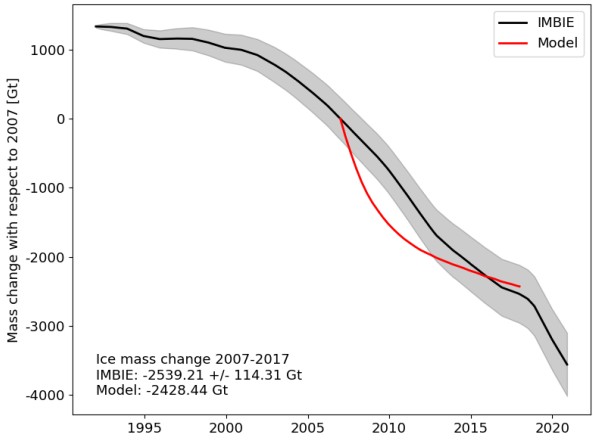

**Figure 2.** Greenland Ice Sheet total ice mass change during the calibration period (2007-2017). The model results (red) are shown next to the Ice Sheet Mass Balance Intercomparison Exercise (IMBIE) estimate (black, Otosaka et al., 2023) and uncertainty range (shaded grey) for comparison.

The goal of this calibration run is to calibrate the calving scheme to be used in our transient simulations, such that the transient tendency of the ice sheet matches recent observations. We use the von Mises calving parameterization from Morlighem

et al. (2016), where calving rate is set as a fraction of ice flow speed at the terminus depending on the local stress state. When local tensile stress is above a stress threshold, $\sigma_{max}$, the calving rate exceeds the local speed of ice flow, and the terminus retreats overall due to calving. We set the calving rate to 0 if the bedrock is above sea-level. We calibrate $\sigma_{max}$ on a glacier-by-glacier basis, and we use two observational constraints for the calibration. First we adjust $\sigma_{max}$ at each marine-terminating glacier, so that simulated retreats match observed retreats along flowlines from Wood et al. (2021) over the calibration period. We use

the same set of tunable glaciers with sufficiently well-constrained bathymetry as the method develop by Choi et al. (2021), corresponding to the 148 glaciers where dynamic ice front motion is simulated, driven by calving rates as well as frontal melting rates (Eqs. 1-2). For glaciers with an ice shelf in North Greenland, we assume that $\sigma_{max}$ of floating ice is 30% of its value for upstream grounded ice, following Åkesson et al. (2022). As a second metric for our calibration, we use the total GrIS mass loss over 2007-2017 from Otosaka et al. (2023), which amounts to 2539±114 Gt. As shown by Goelzer et al. (2020),





most ice sheet models underestimate Greenland mass loss over the recent historical period. Here, in our model calibration, we prioritize matching the observed total mass loss over matching individual glacier retreat rates. Based on our two observational constraints, we establish a simple calibration framework: we target a 2007-2017 GrIS mass loss within observational uncertainties, under the constraint that the simulated retreat values of the 148 marine glaciers are within 1 km of the observed retreat values. All calibration occurs by modifying $\sigma_{max}$ at marine-terminating glaciers to match both constraints at individual glaciers

and the whole ice sheet mass balance.

Figure 2 shows the total ice sheet mass change record from Otosaka et al. (2023), which is compared with the modeled mass loss from our calibration run. The total modeled 2007-2017 mass change agrees with the observational record within uncertainty ranges, even though the modeled mass loss rate is over- and under-estimated in the early and later years of the calibration period, respectively. Figure 3 compares the observed and modeled 2007-2017 retreat rates at the 148 marine glaciers

subject to calibration of $\sigma_{max}$. Our calibration procedure yields a coefficient of determination ($R^2$) of 0.60 and root mean square error of 1.6 km, where these performance metrics are computed over the entire glacier population. While in general we are able to meet our constraint of simulated calving front retreats within 1 km of observed, it is apparent from the RMSE greater than 1 km that we overestimate the retreat of 79-North in North-East Greenland and underestimate the retreat of Sermeq Kujalleq ("SK", previously known by its Danish name Jakobshavn Isbræ). Prior studies have indicated that 79-North may be vulnerable

to large retreats in the near future (Choi et al., 2017) and during the Holocene (Roberts et al., 2024). In our calibration run, this sensitivity of 79-North results from the high sensitivity of the von Mises calving parameterization: this glacier exhibits a threshold-like behavior as it either retreats or advances excessively during calibration for any $\sigma_{max}$ value. We discuss the calibration of SK further in the Discussion section.

## 2.4   Transient runs 2018-2203

We use our statistical models of processed climatic forcing and calibrated ice sheet model configuration to perform long-term simulations of the GrIS. Using StISSM (Verjans et al., 2022), we perform two model large ensemble experiments, as detailed in this section (and listed in Table 1). All simulations start from 2018, i.e., from the final state of the deterministic calibration run, and run until 2203. This simulation period is chosen to align with availability of climate forcing from AWI-ESM simulations (maximizing the length of simulation rather than ending in 2200). We run two deterministic simulations for

each of the scenarios considered, applying the mean climate forcing (pre-2000 control and RCP8.5 emissions scenario), but no internal climate variability. All other simulations include stochastically-generated climatic forcing calibrated to the long-term outputs from AWI-ESM and dEBM, as detailed in Sections 2.1 and 2.2.

### 2.4.1   Pre-2000 control

In order to compare the influence of natural variability versus forced trends on the Greenland Ice Sheet, we perform a control

ensemble (indicated hereafter by the prefix CTRL) with stationary forcing from the AWI-ESM pre-industrial climate. In other words, CTRL ensembles assume no trend in SMB, runoff, or ocean thermal forcing: the mean state, covariance matrix, and internal climate variability statistics are held constant to their 1850-1999 levels. All ensemble members share the same forcing





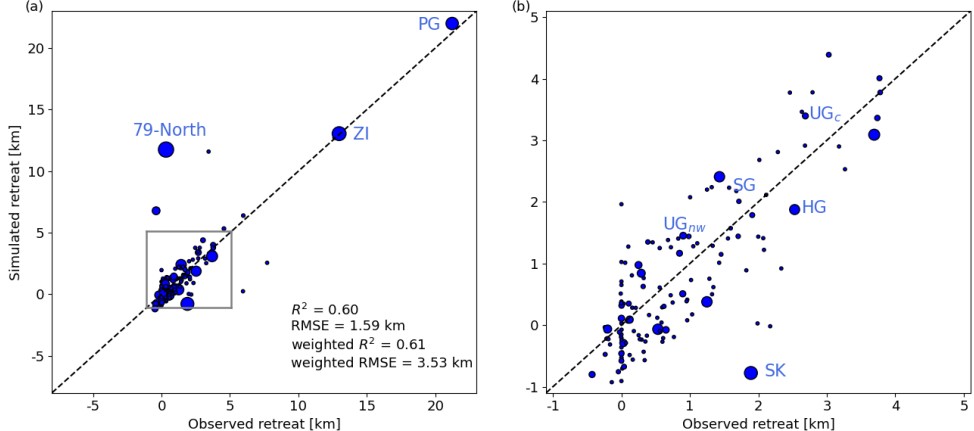

**Figure 3.** (a) Retreat of the 148 calibrated marine glaciers during the calibration period (2007-2017), where the area of each circle is proportional to the catchment size of the glacier it represents. The performance statistics coefficient of determination ($R^2$) and root mean square error (RMSE) are provided unweighted, and weighted by the glacier catchment size. Positive retreat indicates retreat, negative retreat indicates advance. (b) Zoomed-in version of the grey square box shown in (a). Glaciers discussed in the main text are identified, where PG: Petermann Glacier, ZI: Zachariae Isstrom, SK: Sermeq Kujalleq, UG: Upernavik Glacier, SG: Steenstrup Glacier, and HG: Humboldt Glacier. Note that we identify the two main branches of UG: northwestern ($UG_{nw}$) and central ($UG_c$). See Fig. 7a for glacier locations.

statistics. However, they differ by different random realizations of the internal variability component in the climate forcing throughout the simulation. The resulting large pre-2000 control ensemble (CTRL-LE) consists of 100 member simulations.

### 345 2.4.2 High-emissions scenario (RCP8.5)

To quantify the relative importance of forced trends in mean climate to internal variability, we perform an ensemble with changing mean climate forcing following the high-emission RCP8.5 scenario simulation of AWI-ESM (indicated hereafter by the prefix WARM). As described in section 2.2.1, we fit piecewise linear trends separately for each glacier catchment for *TF*, SMB, and runoff forcings with breakpoint in 2000, 2050 and 2100. The changing amplitude of climate variability in this

forced climate model simulation (Fig. 1) also requires us to specify a different covariance matrix for each of these periods (see Appendix A). The resulting large RCP8.5 ensemble (WARM-LE) consists of 100 member simulations.

We briefly note that our choice of the RCP8.5 high emissions scenario is mainly motivated by expediency, with availability of long running climate model simulations, and easy comparison to other ice-sheet model intercomparison projects. The two scenarios considered (CTRL and WARM) are meant to be end members of a broad range of potential future emissions scenarios.

Future work could further investigate this question running large ensembles for other emissions scenarios and other global climate models.



### 2.4.3 Small ensembles

To diagnose the drivers of variability in the simulated GrIS mass change, we perform several additional smaller sub-ensembles with 30 members each, both for the WARM and CTRL forcings (detailed in Table 1). Specifically, we perform stochastic

ensembles with variability only in SMB (OnlySMB-SE), variability only in ocean $TF$ (OnlyOCN-SE), and with variability in all variables but zero covariance between all variables and catchments (NoCOV-SE). The NoCOV-SE applies a diagonal covariance matrix $\Sigma$, i.e., without any correlation between different variables and different catchments, but with the magnitude of variability on the main diagonal identical to that in their corresponding full ensemble. For the WARM ensemble only, we also perform an additional small ensemble (WARM-STN-SE) where the covariance matrix does not change in time, and is

fixed to the covariance matrix of the 2000-2050 period, i.e., the internal climate variability is stationary. This eliminates the increase in amplitude of climate forcing variability in WARM, but the deterministic trends of WARM-STN-SE is identical to the WARM scenario. For the CTRL ensemble, we also performed two additional sets of small "branched" ensembles, which were initialized from a single simulation from the large ensemble in 2032 and 2041 (CTRL-BRN32-SE and CTRL-BRN41-SE, respectively). These small ensembles are designed to elucidate the role of particular events within the simulations in generating

ensemble spread, as will become clear in the Results Section.

| Ensemble Name | Members | Mean Forcing | Variability Included | Start Year |
|---|---|---|---|---|
| CTRL-LE | 100 | Pre-2000 | All | 2018 |
| WARM-LE | 100 | RCP8.5 | All | 2018 |
| CTRL-OnlySMB-SE | 30 | Pre-2000 | SMB | 2018 |
| CTRL-OnlyOCN-SE | 30 | Pre-2000 | *TF* | 2018 |
| CTRL-NoCOV-SE | 30 | Pre-2000 | Uncorrelated | 2018 |
| WARM-OnlySMB-SE | 30 | RCP8.5 | SMB | 2018 |
| WARM-OnlyOCN-SE | 30 | RCP8.5 | *TF* | 2018 |
| WARM-NoCOV-SE | 30 | RCP8.5 | Uncorrelated | 2018 |
| WARM-STN-SE | 30 | RCP8.5 | Stationary | 2018 |
| CTRL-BRN32-SE | 30 | Pre-2000 | All | 2032 |
| CTRL-BRN41-SE | 30 | Pre-2000 | All | 2041 |

**Table 1.** List of ensembles of model experiments discussed in this study, with details on ensemble differences. In the column "Variability Included", "All" refers to SMB, runoff, *TF*, and the correlation between those three variables across Greenland catchments.



## 3 Results

The aim of this study is to investigate the role of internal climate variability in driving mass change from the Greenland Ice Sheet in the future. As a point of comparison, we run two deterministic simulations applying the mean climate forcing for the CTRL and WARM scenarios, but omitting temporal variability in climate forcing. Figure 4a shows the evolution of ice mass for these two simulations as dark blue and dark red dashed lines, respectively. The first 30 years of ice sheet evolution in both deterministic simulations is very similar, indicating that the early evolution is likely driven by a combination of the ice sheet state in 2018 and the pre-2018 mean climate forcing. During these 30 years of similar evolution, there are two notable sub-decadal periods of rapid ice mass loss common to both deterministic simulations (and all other simulations in this study, as we will discuss in the next sections). These rapid ice loss events are associated with the rapid retreats of Petermann Glacier (PG) in the 2030's and Zachariae Isstrom (ZI) in the 2040's. The loss of floating ice at PG in the first 15 years of the simulation triggers a rapid retreat over a relatively flat bed, until the calving front reaches a region of prograde bed topography. The simulated evolution of ZI starting in 2018 is the ongoing response to the complete loss of floating ice between 2004 and 2012 (Khan et al., 2022). The subsequent calving front retreat proceeds over several wide bed bumps until the frontal geometry adjusts to a more gradually sloped configuration. Over the course of the deterministic simulations, 50% and 80% of all marine-terminating glaciers retreat by more than 1 km in the CTRL and WARM deterministic simulations, respectively. However, slower retreat of smaller glaciers contribute less to the overall ice sheet mass loss rate, compared to the cases of PG and ZI.

After ∼2050, the two deterministic simulations diverge. Over the remaining 150 years of the simulation, the CTRL climate forcing causes recovery of approximately half of the ice sheet mass loss from the PG and ZI retreats. In contrast, the WARM deterministic simulation continues to lose mass at a rapid and regular pace, mostly driven by increasingly negative SMB across the ice sheet. The CTRL deterministic simulation contributes to 6.0 cm of sea level rise equivalent in 2100, and 3.1 cm by the end of the simulation in 2203. Ice loss from the WARM deterministic simulation contributes to an equivalent of 11.5 cm of sea level rise in 2100, and 25.4 cm by the end of the simulation in 2203. Though these simulations are not intended as projections, they do fall well within the general range of other Greenland Ice Sheet projections (Goelzer et al., 2020), as discussed in more detail in the Discussion Section.

### 3.1 Control Large Ensemble (CTRL-LE)

The Control Large Ensemble with Pre-2000 climate conditions (CTRL-LE hereafter) includes realistic internal climate forcing variability represented as stochastic temporal variability in SMB, runoff, and *TF* in 100 ensemble members. In Fig. 4a, we show the ice mass evolution of all members of CTRL-LE. The main features of the ice mass evolution in the deterministic control run (rapid retreat of PG and ZI, and gradual recovery after 2050) occur in all members of CTRL-LE. By 2203, the mean ensemble ice loss is 4% greater than in the deterministic simulation, suggestive of noise-induced drift in the ensemble as found in several prior studies of the ice sheet response to climate variability (Tsai et al., 2017; Hoffman et al., 2019; Robel et al., 2024). This difference corresponds to only 1 standard deviation of the ensemble final mass change, and 17 members





**Figure 4.** Ice mass evolution in GrISLENS large ensembles. (a) Ice mass change for all ensemble members (colored lines) and deterministic runs (dashed darker lines). (b) Anomaly of ensemble members with respect to their corresponding deterministic run. (c) Ensemble standard deviation ($\sigma$) over time. (d) Ensemble standard deviation relative to the ensemble absolute mean mass change. Blue lines and shading are CTRL-LE and red lines and shading are WARM-LE. In (b), the shading shows the [5%,95%] range of the ensemble at any time step, and the lines show the 5% members with the lowest and largest ice mass at the final time step. On the right y-axes, mm SLE denotes mm sea-level rise equivalent.

have a smaller ice loss than the deterministic run. We disentangle the mechanisms of this drift more fully in Section 3.3 with comparison to small ensemble experiments.

Figure 4b shows the ice mass evolution of each member as an anomaly with respect to the deterministic run. It is clear that most of the CTRL-LE ice mass anomalies are negative, confirming the 4% larger mass loss from the CTRL-LE ensemble mean compared to its corresponding deterministic run. Another feature apparent in Fig. 4b is that the members at the lowest and highest end of the ensemble mass loss at the final time step (lines) mostly display low-end and high-end ice mass anomalies



throughout the simulation. That is, their ice mass deviation from the ensemble mean is caused by climatic perturbations early

in the simulation that produce a response which persists and continues growing over the simulation period. This persistence is characteristic of dynamical systems involving long response time scales and positive feedback processes. In our simulations, this includes the SMB-elevation feedback and the dynamic thinning propagating upstream following marine glacier retreat.

Figure 4c shows the standard deviation between ensemble members of CTRL-LE (blue line), hereafter referred to as "ensemble spread". The CTRL-LE spread increases until 2150, before leveling off. During the rapid retreat of PG and ZI, the

ensemble spread temporarily increases by 25% for the PG retreat and by almost 300% for the ZI retreat. In both cases, the ensemble spread quickly returns to the trajectory of gradual increase once retreat has occurred in all ensemble members. At the end of the simulation period, the ensemble spread amounts to 1.3 mm of equivalent uncertainty in sea level contribution, and 2.5 mm uncertainty at its peak during the retreat of ZI. To put this ensemble spread in context, in Figure 4d we also show the ratio of the ensemble spread to the ensemble mean ice loss. This ratio represents the relative importance of uncertainty through

the simulation. Over the first 20 years of the simulation period, ensemble spread accounts for 10-300% of the ensemble mean mass loss. Once ZI and PG retreat, the ensemble spread never exceeds 5% of the mean ice loss. However, this relative uncertainty is steadily growing from 2075 as a result of the ice mass recovery of the ensemble mean (Fig. 4a). These results indicate that in the near future (decades), climate variability plays an important role in the progress of Greenland ice mass loss. Once the total mass loss ramps up, climate variability is a less important source of uncertainty on the ice sheet scale compared to the

total ice loss and scenario uncertainty (i.e., the difference between CTRL-LE and WARM-LE as seen in Fig. 4a).

Figure 5 shows the mean (panel a) and spread (panel c) of terminus retreat at all marine-terminating glaciers where calving and melt forcing occur in CTRL-LE. The retreat of some glaciers is highly uncertain, with some ensemble members retreating 10's of kilometers, and others not at all. For most glaciers, the ensemble spread in Fig. 5c-d remains approximately constant after 2050, indicating that most of the uncertainty stems from the internal climate variability causing retreats in some ensemble

members early in the simulations. Figure 6 shows flowline thickness profiles for all CTRL-LE ensemble members at selected glaciers and times. PG (Fig. 6a) and ZI (Fig. 6b) experience the most uncertainty in simulated terminus position in the midst of rapid retreat, with ensemble members differing by up to 10 km, before converging following the main period of retreat. In contrast, at other glaciers, including Upernavik and Steenstrup Glaciers (Fig. 6c,d), the extent of retreat sensitively depends on the details of climate variability. The result is that individual ensemble members undergo retreats that differ by up to 10 km

in extent, and this difference can persist until the end of the simulation period. For these glaciers, there are a discrete number of positions where the front persists in different ensemble members, puncated by rapid retreats on sub-decadal time scales between positions. This is a known behavior of marine-terminating glaciers retreating over bed topography with many local peaks of varying height (Robel et al., 2022). PG and ZI are among the largest catchments in Greenland, so when retreats occur over a sufficiently short time period (less than a decade) and across all ensemble members, small differences in the timing of

retreat onset cause the large increase in total ice sheet mass ensemble spread (Fig. 4c). Other glaciers are either sufficiently small or retreating relatively slowly, and so even when there are persistent and large differences in the extent of retreat (e.g., at Upernavik and Steenstrup), they have limited impact on the ensemble mean or spread.



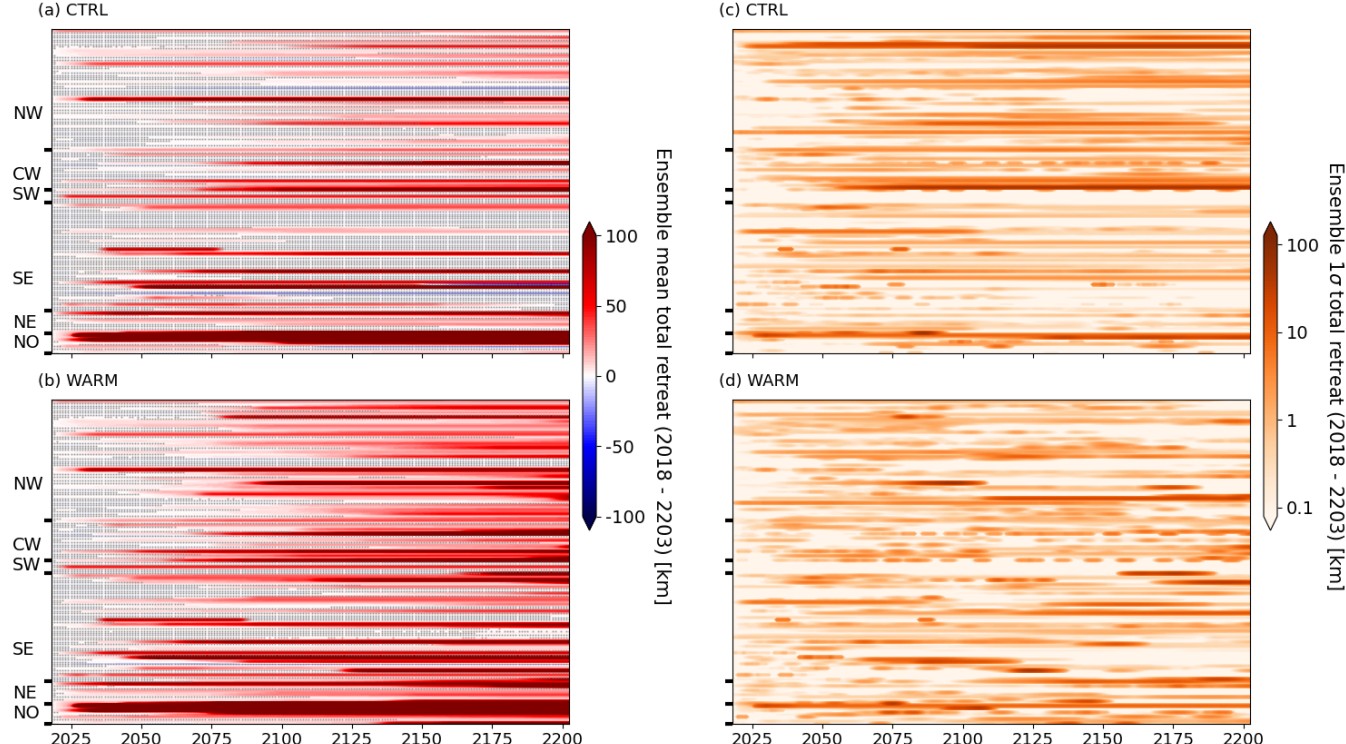

**Figure 5.** Terminus retreat for each marine-terminating glacier in Greenland, where glaciers are grouped by regions following NO: North, NE: North-East, SE: South-East, SW: South-West, CW: Central-West, and NW: North-West. (a-b) Ensemble mean of retreat extent [km] in (a) CTRL-LE and (b) WARM-LE. Red indicates retreat, blue indicates advance. (c-d) Ensemble spread of retreat extent [km] in (c) CTRL-LE and (d) WARM-LE. In (a,b), grey hatching denotes non-significant retreat/advance, where significance is evaluated as agreement on the sign of retreat between 95% of ensemble members. Note the logarithmic colorbar in (c), (d).

Thickness changes in CTRL-LE are mapped in Fig. 7, with the top row showing ensemble-mean thickness changes at 2050, 2100 and 2203, and the bottom row showing corresponding ensemble spread. Before 2100, a dynamic thinning of hundreds

of meters occurs far upstream of retreating termini at PG, ZI and other marine-terminating glaciers in West and Southeast Greenland. Gradual thickening of tens of meters occurs across the ice sheet interior, mainly after 2100. This progression largely explains the ice mass evolution in Fig. 4a, with dynamic thinning driving most ice loss early in the simulation, and positive SMB in the ice sheet interior causing thickening later.

During the retreats of PG and ZI, slight differences in retreat timing between ensemble members causes tens to hundreds of

meters of uncertainty in the upstream-propagating wave of dynamic thinning (e.g., at ZI in Fig. 7d), which quickly dissipate once all ensemble members are retreated (Fig. 7e,f). In contrast, at glaciers where uncertainty in retreat extent persists (e.g., Upernavik Glacier), there remains hundreds of meters of uncertainty in dynamic thinning, which gradually spreads upstream and into adjacent catchments over the simulation period (Fig. 7e,f). SK does not begin to undergo retreat until after 2050 in




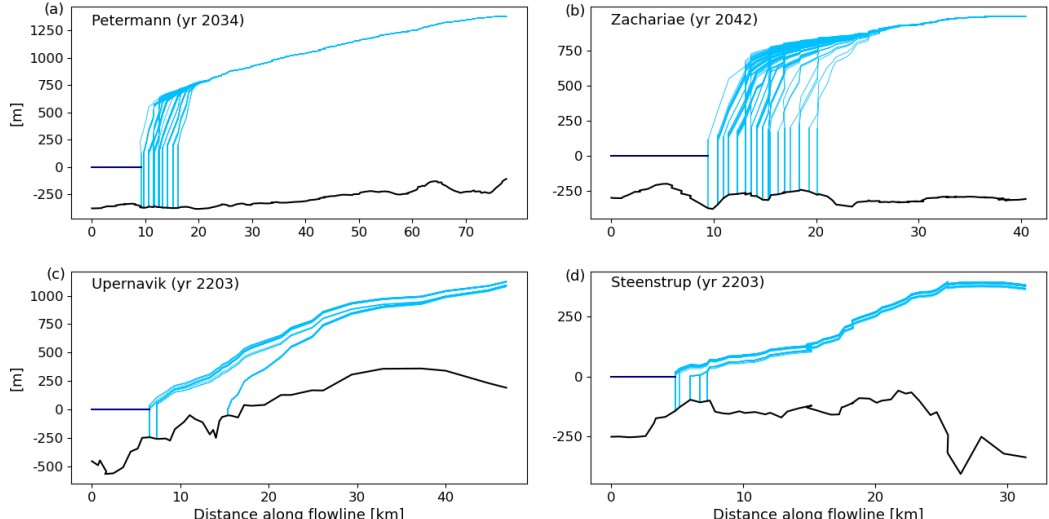

**Figure 6.** Along-flow profiles of ensemble near-terminus ice thickness for 4 glaciers at various points during simulation period in CTRL-LE. (a) Petermann Glacier (year 2034). (b) Zachariae Isstrom (year 2042). (c) Upernavik Glacier (year 2203). (d) Steenstrup Glacier (year 2203). All 100 ensemble members are included in the plots and each profile corresponds to single ensemble member. See Fig. 7a for glacier locations.

CTRL-LE, though differences among ensemble members in retreat onset timing produce large uncertainties in thinning during

the middle part of the simulation period (2070-2150, see Fig. 7e). By the end of the simulation, SK has retreated in all ensemble members, and thinning uncertainty has largely dissipated (Fig. 7f). Across glacier catchments in the Southern half of Greenland, SMB variability drives gradually increasing uncertainties in thickness evolution over large areas; it generally remains an order of magnitude lower than that of retreating glaciers in western Greenland (Fig. 7 d,e,f). The relative importance of ocean-driven dynamic thinning and SMB variability is discussed in further detail in section 3.3.





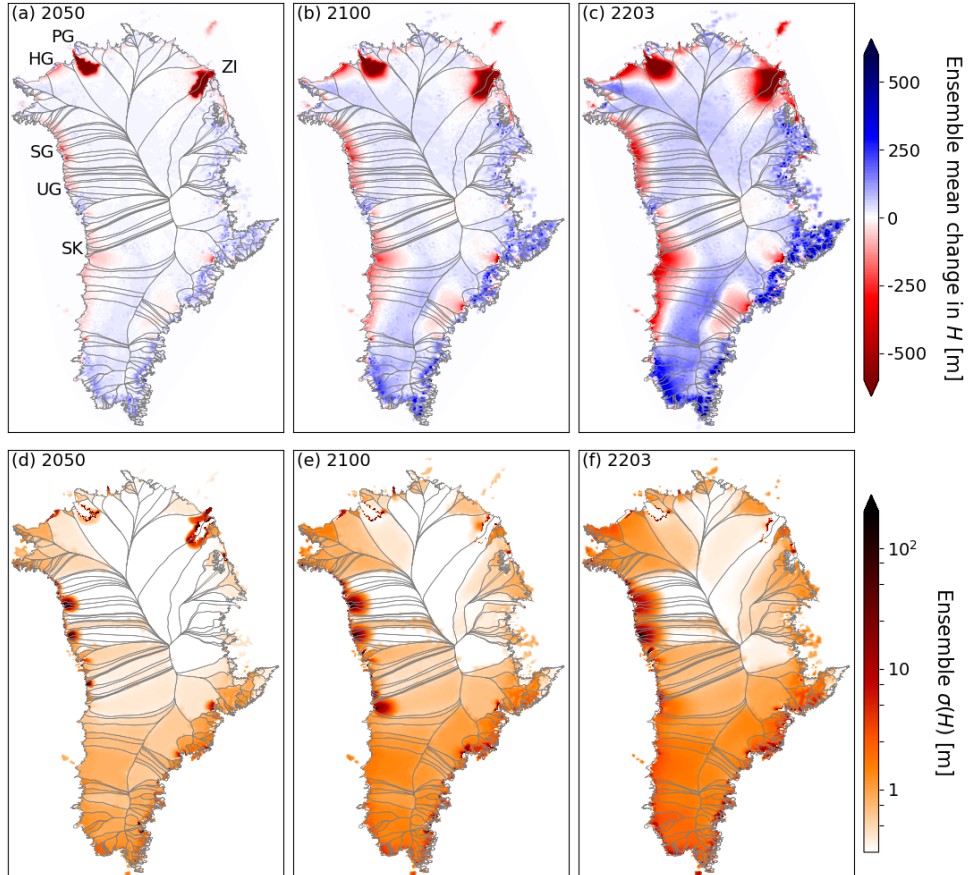

**Figure 7.** Ice thickness ($H$) in CTRL-LE. Top row: ensemble mean change in $H$ between 2018 and (a) 2050, (b) 2100, and (c) 2203. Bottom row: ensemble spread ($\sigma(H)$) in (d) 2050, (e) 2100, and (f) 2203. Note the logarithmic color scale in (d,e,f). Color scales are identical to those in Fig. 8. Glacier locations are shown in (a) for PG: Petermann Glacier, ZI: Zachariae Isstrom, SK: Sermeq Kujalleq, UG: Upernavik Glacier, SG: Steenstrup Glacier, and HG: Humboldt Glacier.

## 3.2 RCP8.5 Large Ensemble (WARM-LE)

The RCP8.5 large ensemble (hereafter WARM-LE) is initialized identically to CTRL-LE, but the statistics of the climate forcing evolves over time, leading to increasingly intense surface melt and ocean thermal forcing over the course of the simulation period. WARM-LE is meant to be a point of comparison with CTRL-LE. First, we determine how the ice sheet responds differently to changes in the mean climate, as compared to internal climate variability. Second, we analyze how the response to internal variability may differ under different background climatic states. We note briefly that RCP8.5 is a high-end-member emissions scenario and the following results should be interpreted as a model experiment, not a projection.

The evolution of the WARM-LE ice mass (Fig. 4, red lines) is qualitatively similar to the deterministic simulation (dashed dark red line). At the end of the simulation period, spread in WARM-LE ice mass loss accounts for 2.4 mm of equivalent





uncertainty in sea level rise, as compared to the ensemble-mean ice mass loss equivalent to 255 mm sea level rise. Prior to
2050, the WARM-LE mean and spread (Fig. 4a,c) track very closely to CTRL-LE, as the ice sheet response is dominated by
retreat of PG and ZI. This similarity confirms that the behavior early in the simulation is largely caused by a combination of
the initial ice sheet state and the ice sheet disequilibrium with the climate at the beginning of the simulation. Evolving mean
climate forcing (Fig. F1) appears to play little role in these first few decades of the ensemble. Prior to the retreat of PG, the
ensemble-mean ice mass loss is sufficiently higher in WARM-LE than in CTRL-LE, that the ratio of ensemble spread to mean
ice loss (Fig. 4d) is generally lower, though still substantial (10%-30%) in the first ~20 years of the simulation period. This
effect is also evident on longer time scales, where all ensemble members have a common ice sheet response to increasingly
intense surface melt and ocean thermal forcing that dominates differences between ensemble members caused by internal
climate variability. Finally, Fig. 4b shows the anomaly of each WARM-LE member with respect to the deterministic WARM

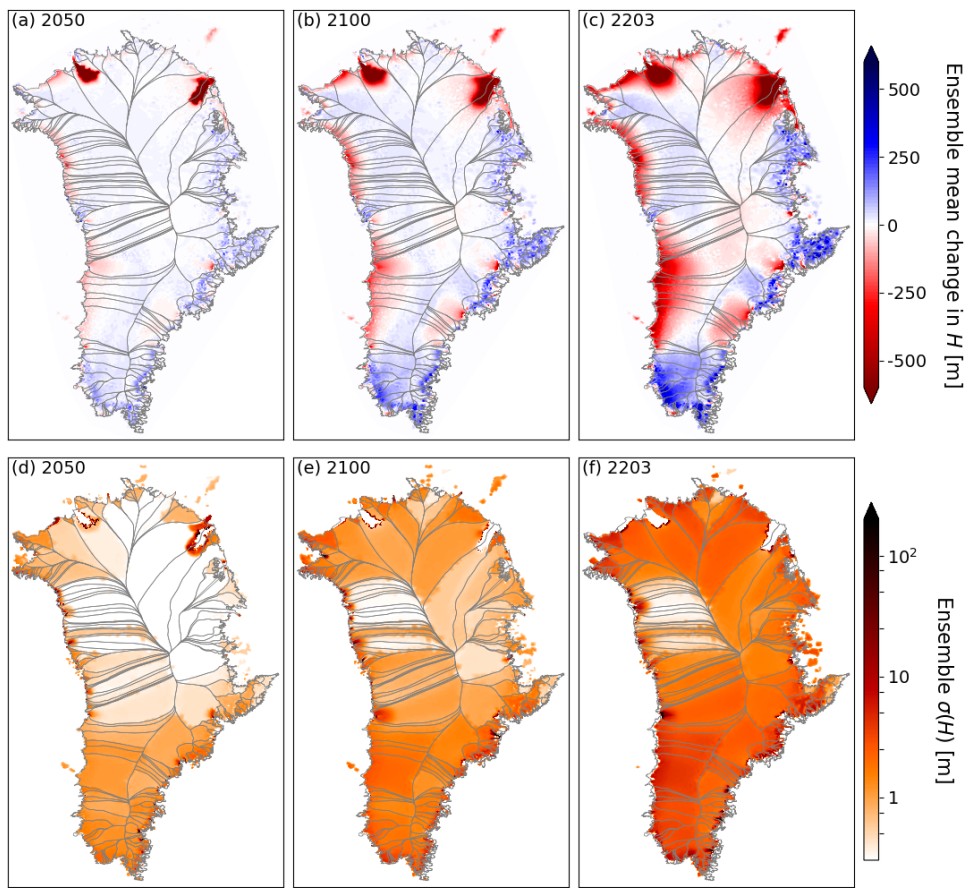

**Figure 8.** Ice thickness ($H$) in WARM-LE. Top row: ensemble mean change in $H$ between 2017 and (a) 2050, (b) 2100, and (c) 2203. Bottom row: ensemble spread ($\sigma(H)$) in (d) 2050, (e) 2100, and (f) 2203. Note the logarithmic color scale in (d,e,f). Color scales are identical to those in Fig. 7.




run. Similarly to CTRL-LE, there is a pronounced persistence of early stochastic perturbations to the ice sheet mass through
480 the entire simulation period. As a result, the members with the most extreme final ice mass totals are generally characterized
by trajectories being at the lower or upper end of the ensemble already in the first few decades, or even years (Fig. 4b).

Figure 8 shows the evolution of ensemble mean and spread in ice thickness for WARM-LE, and Fig. 9 shows the difference
between WARM-LE and CTRL-LE in terms of ensemble mean and spread in thickness at the end of the simulation period. In
the first few decades of the simulation period, several marine-glaciers in West Greenland undergo rapid retreat and dynamic
485 thinning (Fig. 8a and glaciers numbered ~72-130 in Fig. 5c). One major difference with CTRL-LE is that these retreats are
more uniform across ensemble members (compare glaciers numbered ~72-130 in Fig. 5b and 5d). This results in persistently
less ensemble spread in near-margin thinning in these West-Greenland catchments (Fig. 9b). In contrast, throughout the sim-
ulation period, there is greater ensemble spread in the catchment-wide thickness across most of the Greenland interior. Such
a catchment-wide response is caused by variability in SMB. An increase in the amplitude of SMB variability (discussed in
490 more detail in the next section) is already apparent early in the simulation period and continues to drive growing spread in
thickness until the end of the simulation period. The greater areal extent of this SMB-driven increase in ensemble spread drives
the overall greater spread in ice mass in the later part of the WARM-LE simulation period (Figure 4b).

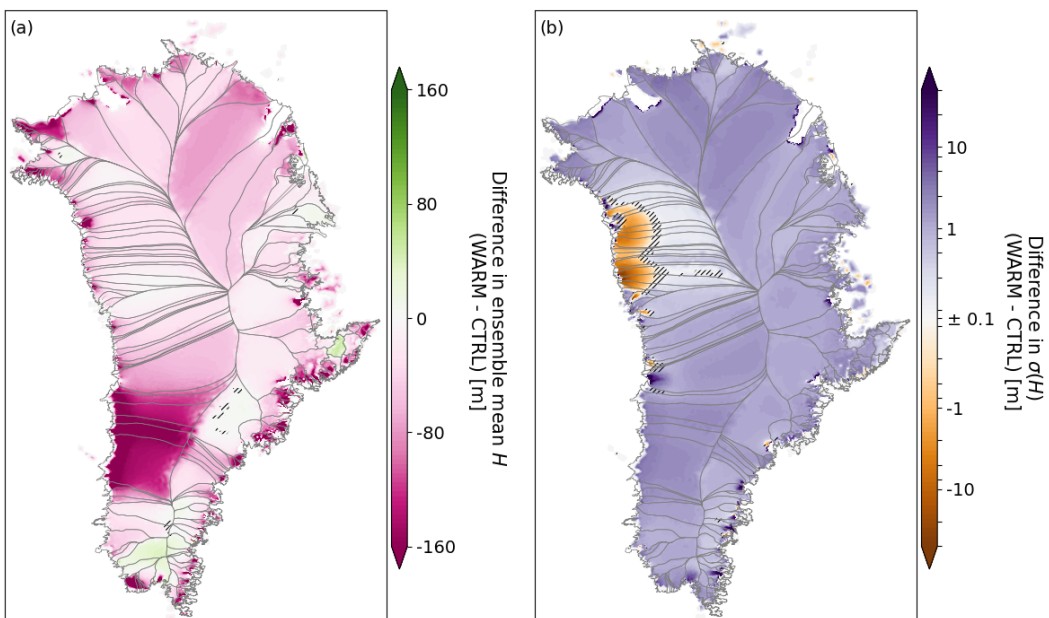

**Figure 9.** Difference in ice thickness change in 2203 between CTRL-LE and WARM-LE in terms of (a) ensemble mean (pink is greater
ensemble-mean thinning in WARM-LE), and (b) ensemble spread ($\sigma(H)$, blue is more ensemble spread in WARM-LE). Note the pseudo-
logarithmic color scale in (b). Hatching denotes non-significant differences, evaluated with a 300-sample bootstrap procedure and controlling
for a false discovery rate of 0.05 (see Appendix G).



After 2050, ice sheet evolution in WARM-LE departs more strongly from CTRL-LE (compare Figs. 8b,c and 7b,c, and see Fig. 9a). By this point in the simulation period, the mean climate forcing has strongly diverged between the two large ensembles (Fig. F1), and the ice sheet has had sufficient time to respond to the different climate forcing. SMB has decreased enough to cause thinning across almost the entire ice sheet interior except for some land-terminating glacier catchments in far South Greenland (Fig. 8b,c). Marine-terminating glaciers across the whole ice sheet experience more extensive retreat and more intense dynamic thinning, particularly between 2100-2200 (Fig. 8c). WARM-LE spread in ice thickness is higher than CTRL-LE across most of the ice margins (Fig. 9b), indicating that retreat and the resultant dynamic thinning are variable across ensemble members, driven by variability in mean surface melt in the ablation zone and ocean melt at calving fronts. In all WARM-LE simulations, SK undergoes an early retreat that does not occur until later in CTRL-LE. However, at the end of our simulation period, the strong ocean forcing has driven a second retreat at SK, which drives an increase in ensemble spread. It may be that given further simulation time, this could grow to become a substantial source of ensemble spread. WARM-LE spread in ice thickness is also higher than CTRL-LE across the ice sheet interior, though to lesser extent than at the margins, likely resulting from the increasing amplitude of SMB variability through the simulation period.

### 3.3 Small Ensembles

Our explicitly stochastic approach to representing internal variability of climate forcing in ice sheet simulations allows us to systemically remove or modify different sources of climate variability in our simulations. To determine which aspects of climate variability are the primary contributors to the leading-order statistical moments and certain behaviors in the large ensembles discussed in the prior sections, we analyze a series of "small" ensembles (described in Sect. 2.4.3). For both CTRL and WARM climate forcing, this includes OnlySMB-SE, OnlyOCN-SE, and NoCOV-SE. In addition, the small ensembles include WARM-STN-SE, i.e. the WARM scenario with fixed covariance matrix, and the two branched CTRL-BRN32-SE, and CTRL-BRN41-SE, branching from the CTRL scenario in 2032 and 2041. In light of Fig. 4, the purpose of the two latter small ensembles is clear: by branching the ensemble at these years, we eliminate all pre-existing ensemble spread in order to understand the role of the retreats of PG (starting 2033 in most LE members) and ZI (starting 2042 in most LE members) in generating ensemble spread.

### 3.3.1 Control Small Ensembles

In Fig. 10a, we show the distribution of ice mass change across the large and small ensembles with CTRL climate forcing. In general, we find that all aspects of variability contribute somewhat to the large ensemble spread, including SMB, ocean, and the covariance structure of the climate forcing. Omitting covariance between SMB and ocean thermal forcing variability and between catchments (CTRL-NoCOV-SE; dark red data in Fig. 10a) reduces ensemble spread by 44%. Applying SMB variability only (CTRL-OnlySMB-SE; orange data in Fig. 10a) reduces ensemble spread by 36% and shifts the ensemble towards less ice mass loss, with the mean and median being approximately equivalent to the 75th percentile in CTRL-LE. Applying ocean variability only (CTRL-OnlyOCN-SE; blue data in Fig. 10a) reduces ensemble spread by 61% without a substantial shift in the mean.



Figure 10a also shows that at the end of the simulation period, CTRL-LE (green data) is skewed towards more mass loss (see also Fig. E1 in Appendix). This is indicated visually by a "long tail" of five ensemble members in which there is much more mass loss than the equivalent ensemble members on the other end of the distribution (less mass loss). While prior studies (Robel et al., 2019) have indicated that such high-end skew in simulations of mass loss may occur during periods of rapid

retreat, there are no ongoing rapid retreats of large catchments in CTRL-LE at the end of the simulation period. The time-dependent calculation of skewness (Fig. E1 in Appendix) indicates that this persistent skewness only arises in the last 50 years of the simulation period. Since none of the small ensembles have such skew, we may conclude that correlation between ocean and SMB variability is the cause of this skew, as this is the only factor omitted from all three ensembles (CTRL-NoOCN-SE and CTRL-NoSMB-SE implicitly omit these correlations by zeroing out variability in one or the other type of variability). However,

our small ensembles are too small to robustly attribute skewness on the basis of just a few "outlier" ensemble simulation, thus we leave this possibility as a hypothesis to be further evaluated in future work.

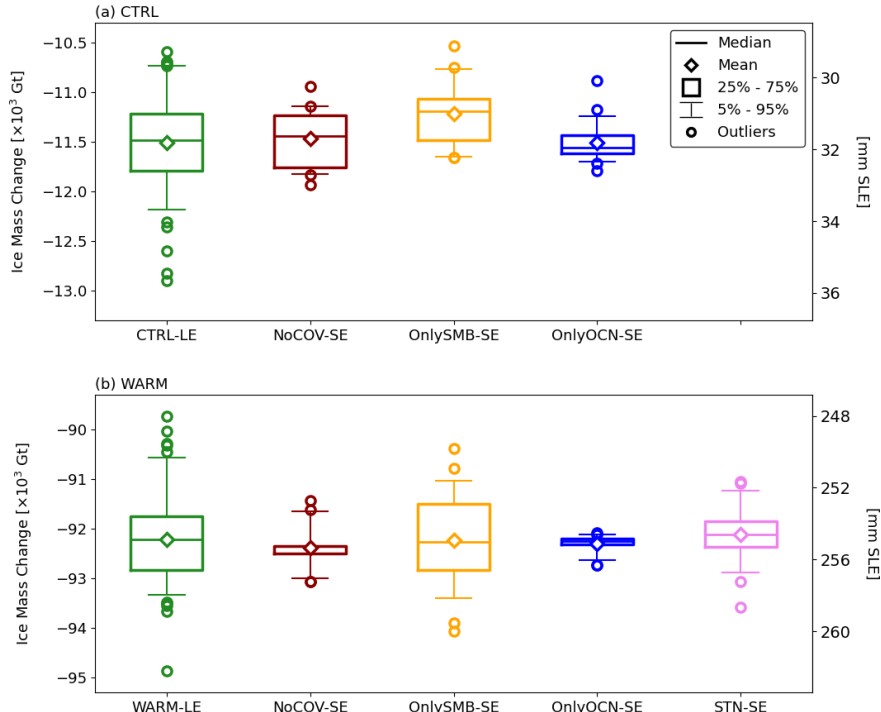

**Figure 10.** Boxplots showing distribution of ice mass change in year 2203 in all ensemble members for the (a) CTRL, and (b) WARM climate forcings. Shown in (a) are CTRL-LE (green), CTRL-NoCOV-SE (dark red), CTRL-OnlySMB-SE (orange), and CTRL-OnlyOCN-SE (blue). Shown in (b) are WARM-LE (green), WARM-NoCOV-SE (dark red), WARM-OnlySMB-SE (orange), WARM-OnlyOCN-SE (blue), and WARM-STN-SE (violet). Horizontal lines indicate ensemble median, diamonds ensemble mean, boxes inter-quartile (25% - 75%) range, whiskers 5% - 95% range, and circles individual outliers. On the right y-axis, mm SLE denotes mm sea-level rise equivalent. Note that the span of the y-axis in (a) is exactly half of that in (b).





These small ensemble results indicate that over the simulation period, the largest drivers of spread in the large ensemble are SMB variability and the spatial covariance between different climate forcing variables and glacier catchments. In Fig. 11a, we show the difference in ensemble spread in ice thickness between CTRL-OnlyOCN-SE and CTRL-OnlySMB-SE for 2203.

SMB variability drives most variability in ice thickness over the Greenland interior. This SMB-driven thickness variability is higher by up to ten times in the South compared to SMB-driven thickness variability in Central and North Greenland. In comparison, ocean variability drives an even higher amplitude of thickness variability over small regions encompassing the dynamic thinning wave from retreat of glaciers, mostly in West Greenland (e.g., Upernavik and SK).

In quantitative terms, we find that over less than 10% of the area of the Greenland Ice Sheet are dominated by ocean-driven

thickness variability, with thickness variability in the rest of the ice sheet being driven by SMB. However, the higher intensity of ocean-driven thickness variability in these small regions means that ocean variability alone (i.e., CTRL-OnlyOCN-SE) drive 40% of the mass balance variability for all of Greenland as simulated in the large ensemble (CTRL-LE).

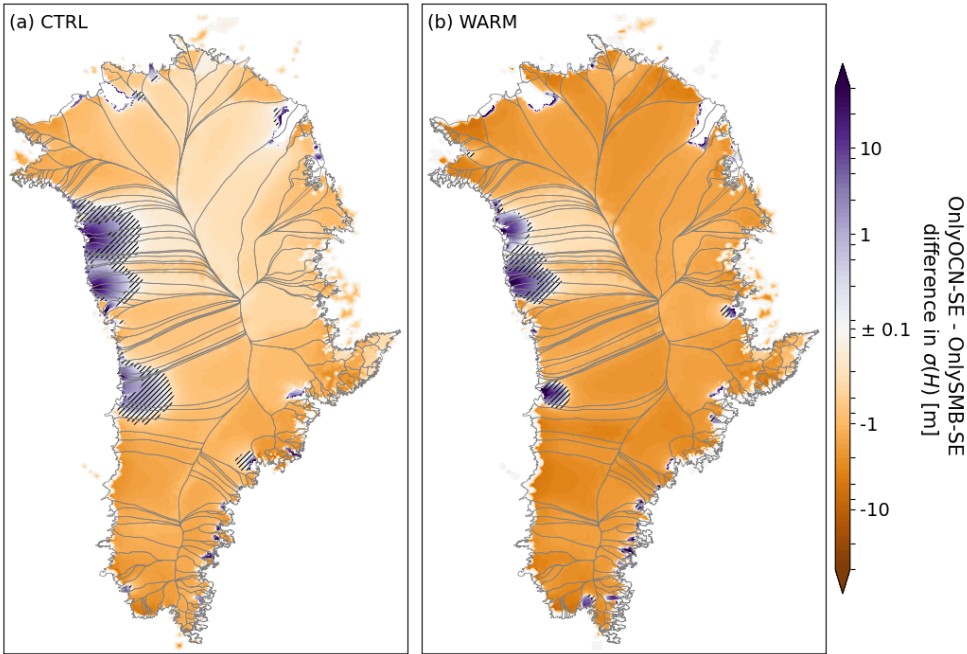

**Figure 11.** Difference between ensemble spread in ice thickness ($\sigma(H)$) in 2203 in OnlyOCN-SE versus OnlySMB-SE for (a) CTRL climate forcing, and (b) WARM climate forcing. Greater spread in OnlyOCN-SE is indicated by blue and greater spread in OnlySMB-SE is indicated by orange. Note the pseudo-logarithmic color scale. Hatching denotes non-significant differences, evaluated with a 300-sample bootstrap procedure and controlling for a false discovery rate of 0.05 (see Appendix G).

If the ensemble mean of the ice sheet mass loss were driven entirely by the climate forcing scenario (i.e., the difference between CTRL and WARM), then we should expect no statistically significant difference between the small ensemble means

of CTRL and the CTRL-LE mean, since only the variability in climate forcing is modified in the small ensembles. We use





a Welch's t-test with unequal variances and sample sizes (Welch, 1947) to determine whether the mean ice mass loss of the small ensembles are significantly different from the CTRL-LE mean. Indeed, there is no significant difference for the CTRL-OnlyOCN-SE and CTRL-NoCOV-SE ($p > 0.5$). However, CTRL-OnlySMB-SE has a shift in the ensemble mean ice loss that is statistically significant ($p < 10^{-3}$), as excluding ocean forcing variability decreases total ice loss. This difference strengthens

the hypothesis of noise-induced drift in CTRL-LE compared to the deterministic run (4% greater mass loss, see Fig. 4a,b), since the deterministic run also lacks ocean variability. This result is consistent with prior work finding that ocean variability drives real drift in the direction of retreat in ice sheet evolution (Verjans et al., 2022; Robel et al., 2024). Since ocean variability drives uncertainty in the onset time of glacier retreat over peaks in bed topography, the model drift here is thus probably entirely caused by noise-induced bifurcations, as described previously by Robel et al. (2024).

### 3.3.2   Branched Small Ensembles

As discussed in previous sections, the rapid retreats of PG and ZI play an important role in generating ensemble spread during the period of the retreats. The branched small ensembles (CTRL-BRN32-SE and CTRL-BRN41-SE) allow us to understand the sources and persistence of this retreat-mediated uncertainty. Figure 12 shows the ensemble spread of total ice mass over the simulation period for CTRL-LE (green line), CTRL-BRN32-SE (dark red line) and CTRL-BRN41-SE (orange line). CTRL-

BRN32-SE shows a similar increase in ensemble spread during the retreat of PG as compared to CTRL-LE, indicating that most of the ensemble spread generated at PG is related to climate forcing variability during the PG retreat driving differences in retreat rate. In contrast, when there is less ensemble spread at the onset of ZI retreat, there are much smaller increases in ensemble spread during the ZI retreat. This indicates the importance of retreat onset timing at ZI to determining the how much the ensemble spread increase during the retreat. Finally, despite substantial differences in the magnitude of ensemble

spread increase during large retreats in the branched small ensembles, the final differences in ensemble spread are similar in magnitude to the differences existing before the retreats simply by starting simulations at a new time. Thus, we conclude that while ensemble spread increases substantially during retreats, this spread does not persist if all ensemble members ultimately undergo the same retreat. This contrasts with the ensemble behavior for glaciers such as Upernavik and Steenstrup Glacier, where some members of CTRL-LE and WARM-LE undergo retreat, while others do not. In such cases, uncertainty is likely to

be persistent.

### 3.3.3   RCP8.5 Small Ensembles

By the end of the simulation period, WARM-LE has about twice as much ensemble spread as CTRL-LE, though the drivers of this spread are not qualitatively different (Fig. 10b). When only SMB variability is imposed (OnlySMB-SE as orange data in Fig. 10b), there is no significant change in the ensemble spread ($p > 0.4$ in Levene's test, Levene, 1960). However, omitting

spatial covariance in SMB between catchments (NoCOV-SE as dark red in Fig. 10b) or SMB variability entirely (OnlyOCN-SE as blue in Fig. 10b) causes ensemble spread to be significantly reduced (54% and 81% reductions, respectively). In WARM-LE, the amplitude of climate forcing variability changes over the simulation period, and we change the prescribed covariance to capture this non-stationarity (e.g., Fig. 1). This is particularly relevant to SMB variability, which amplifies strongly in the



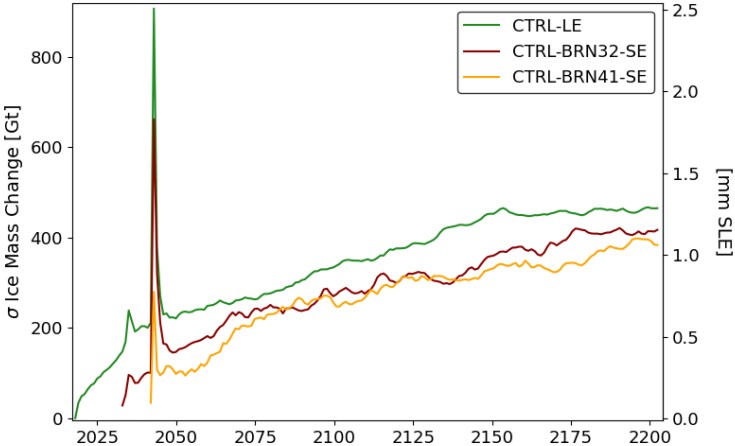

**Figure 12.** Ensemble spread ($\sigma$) in total ice sheet mass change for CTRL-LE (green), CTRL-BRN32-SE (dark red), CTRL-BRN41-SE (orange). On the right y-axis, mm SLE denotes mm sea-level rise equivalent.

AWI-ESM climate model response to the RCP8.5 scenario. As a result, Fig. 11b shows that SMB-driven variability is more
pronounced across Greenland than in the CTRL forcing scenario, and that areas dominated by ocean-driven variability are more restricted. In WARM-STN-SE, we omit the increasing amplitude of variability in SMB, as well as all other climatic forcings, by keeping the forcing covariance constant at 2000-2050 values over the simulation period. As a result, the ensemble spread in WARM-STN-SE is reduced by 33% (violet data in Fig. 10b). This difference is a large fraction of the inter-ensemble spread between WARM-LE and CTRL-LE (33% versus 46%). To investigate this contribution, Fig. 13 shows the ratio of the
interannual SMB variability from dEBM in the last 3 decades of the simulation period to that in the first 3 decades. In most of East Greenland and some catchments in central West Greenland, the amplitude of SMB variability is 2-8 times higher in the final decades of the simulation period, as compared to the starting decades. No catchment shows any appreciable decrease in SMB variability over this period. The WARM small ensembles thus tell a consistent story, that in a strongly warming climate: (1) the full-ensemble spread can be explained by SMB variability alone, and (2) almost 1/3[rd] of ensemble spread at year 2203
can be attributed to the warming-driven increase in the amplitude of SMB variability over the next two centuries.

If there is any drift in the ensemble-mean ice loss in WARM-LE, it is not statistically distinguishable from the larger envelope of ensemble spread in this ensemble at the end of the simulation period. In addition, none of the small ensemble results suggest that a particular variability component causes noise-induced drift, in contrast to the CTRL forcing case (Fig. 10). Furthermore, though there are some outlier ensemble members in WARM-LE, their number is limited, making it difficult to draw conclusions
about skewness of ice mass without a larger ensemble (which would be computationally challenging to run within the context of this study).





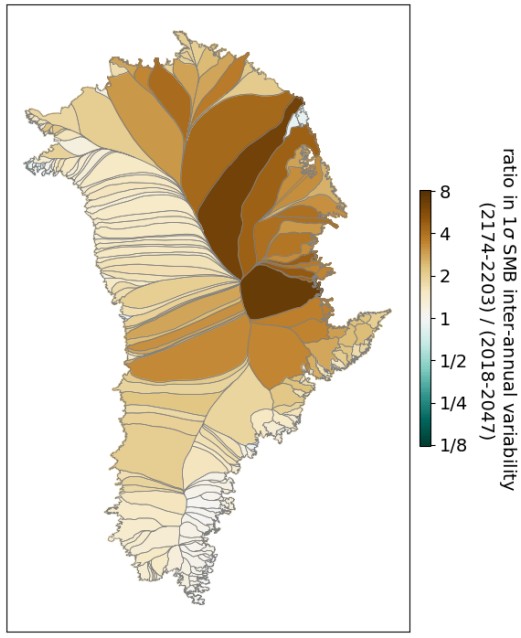

**Figure 13.** Ratio of amplitude of catchment-averaged SMB interannual variability in dEBM under RCP8.5 scenario between the last three and first three decades of our simulations (2174-2203 to 2018-2047). Note that this time-evolving SMB field of dEBM is used to constrain the SMB statistics of the WARM climate forcing.

## 4 Discussion

The main conclusion of this study is that, in the first two decades of simulations, climate variability plays an important role in generating uncertainty in the evolution of total ice sheet mass and contribution to sea level rise. Beyond this initial period, internal climate variability quickly diminishes in importance at the ice sheet scale, particularly when compared to uncertainty in future emissions scenarios. This conclusion derives from a large ice sheet model ensemble constituting the most advanced quantification of the role of climate variability in future Greenland Ice Sheet evolution. The modeling experiments described in this study are unprecedented in terms of: the resolution of ice sheet model simulations (<1 km at calving fronts), the inclusion of dynamic calving, the size of ensembles, and the correction of stochastic climate variability to match the statistics of observations.

Our findings are consistent with broader climate modeling results on key variables such as global and regional trends in temperature and precipitation, where internal variability is a major component of uncertainty in the near future (Lehner et al., 2020). The uncertainty in future Greenland Ice Sheet evolution is largely driven by SMB, spatial correlations in SMB, and the increasing amplitude of variability in SMB. Within individual glacier catchments and time periods of the simulation, uncertainties in mass loss can be considerably greater, due to ocean variability driving uncertainty in the timing of rapid glacier





retreats. This is particularly notable in West Greenland, where individual ensemble members exhibit retreats differing in timing by decades, for example at SK, Upernavik, and Steenstrup Glaciers.

## 4.1 Comparison to prior LE studies

A previous study by Tsai et al. (2017), found that ensemble-mean ice sheet response to the coupled CESM large ensemble is
about 12 cm sea-level rise equivalent in 2100, which is very similar to our simulated WARM-LE ice loss. However, they find an ensemble spread at this time of ∼22 mm, which is approximately one order of magnitude more spread than our value of 1.3 mm at 2100 in WARM-LE. Tsai et al. (2017) is the closest point of comparison for GrISLENS. That study used two large ensembles of a global climate model to force an ice sheet model until 2100 under an RCP8.5 scenario, in order to quantify the role of internal climate variability in driving uncertainty in future Greenland ice loss. The mean climate forcing was bias corrected
to modern observations, but the climate variability was purely as predicted by their global climate model, in contrast to our methods for generating climate variability which are corrected to observed variability (Verjans et al., 2023; Ultee et al., 2024). Additionally, the computational constraints of running climate model large ensembles requires coarse horizontal resolution: $3.75°$ for the ocean-atmosphere coupled LE used by Tsai et al. (2017), and $1°$ for the model in Tsai et al. (2017), or about 3 and 10 grid points across Greenland, respectively. Similarly, the ice sheet model for their study was run at 20 km resolution,
which is too coarse to explicitly resolve most Greenland outlet glaciers. Their ice sheet model was also initialized at a steady state in 2000, which does not capture the strong Greenland mass loss trend, estimated to have started earlier in the 20th century (Mouginot et al., 2019).

Both GRISLENS and Tsai et al. (2017) attribute most of the simulated ensemble spread to SMB variability. Though it is challenging to definitely attribute the cause for these starkly different ensemble spreads, the most likely explanation is the
coarse resolution of the climate model forcing in Tsai et al. (2017) ($\sim 400$ km for the coupled LE forcing compared to 5 km for dEBM, see section 2.1.2). Such a resolution introduces spurious spatial correlation between catchments in SMB, which as we have shown, plays an important role in driving ensemble spread under warming. Indeed, we find that accounting for realistic inter-catchment correlation in WARM-LE more than doubles the ensemble spread compared to assuming uncorrelated catchments. It is therefore unsurprising that representing SMB over Greenland as only ∼10 grid points can cause a further
ten-fold increase in ensemble spread, as this approach assumes almost perfect correlation across large portions of the entire ice sheet.

Tsai et al. (2017) also find substantial evidence of drift between simulations forced with and without internal climate variability, even under strong emissions forcing, which they attribute to the use of a positive-degree day (PDD) scheme for calculating SMB. This is consistent with recent work showing that nonlinearities in PDD schemes (Lauritzen et al., 2023) or other ice sheet
processes (Robel et al., 2024) can lead to noise-induced drift, particularly in models starting from a deterministic steady-state. We directly generate Gaussian stochastic variability in SMB (as indicated by observations; Ultee et al., 2024) and start from a more realistic out-of-balance initial ice sheet state. These factors are consistent with the absence of statistically significant drift in WARM-LE.



## 4.2 Comparison to ISMIP6

We find that aleatoric uncertainty due to internal climate variability is 1-2 orders of magnitude less than structural uncertainty due to ice sheet model differences or uncertainties in the climate model response to emission forcing as quantified in the most recent iteration of the Ice Sheet Model Intercomparison Project (ISMIP6). ISMIP6 is a community effort to compare simulations of ice sheet change for Antarctica (Seroussi et al., 2020) and Greenland (Goelzer et al., 2020). This is generally considered to be the most comprehensive estimate of uncertainties in future ice sheet evolution as it includes differences

between ice sheet models and between climate models. Part of the purpose of our study is to supplement ISMIP6 by designing an ensemble to quantify uncertainty not captured by the ISMIP6 ensemble. ISMIP6 ensemble spread across model simulations of the Greenland Ice Sheet (holding climate forcing constant) ranges from 8-24 mm SLE in 2100. ISMIP6 spread across climate models (holding the ice sheet model constant) ranges from 21-37 mm SLE in 2100. In comparison, in 2100, ensemble spread in WARM-LE is 1.3 mm SLE (or 2.5 and 2.4 mm at the peak of rapid retreats and the end of the simulation period, respectively).

In Fig. 14, we show the difference in ensemble spread among ice sheet models in thickness at 2100 for the MIROC5-RCP8.5 ISMIP6 simulations compared to the WARM-LE ensemble. Across the ice sheet margins, inter-model ensemble spread is generally 1-2 orders of magnitude greater than ensemble spread in WARM-LE. The only places where ensemble spread is greater in WARM-LE than in the ISMIP6 ensemble are in the ice sheet interior, where our methodology samples internal variability in snowfall, compared to the structural uncertainty sampled by using different SMB models in ISMIP6. However, if

future iterations of ISMIP require models to dynamically match historical ice loss transients, this may sufficiently reduce the spread among ice sheet models, that it becomes more comparable to aleatoric uncertainty. The results of this study strongly argue for the value of such an approach.

In the ISMIP6 protocol, all simulation results are presented as differences from a "control" where SMB and ocean thermal forcing are held constant at 2014 values. In ISMIP6, ice sheet models demonstrate a diverse range of responses in this "control"

simulation, with most models having less than 6 mm SLE mass gain or loss by 2100, and all ice sheet models simulating between 50 mm SLE mass gain and 15 mm SLE mass loss (with the 50 mm gain model being a strong outlier) (Goelzer et al., 2020). By comparison, in our study, CTRL-LE has ~50 mm SLR mass loss by 2100. This difference is the result of our initialization procedure which calibrates the model in order to reproduce the observed mass loss for the ice sheet between 2007 and 2017 (Fig. 2). While some ISMIP6 models do reproduce the historical mass loss trend for Greenland, this transient mass

loss trend does not continue past the initialization of future simulations in 2014. This is likely due to most ISMIP6 models using a retreat parametrization whose boundaries would not move under fixed ocean thermal forcing, in contrast to the moving margins of our CTRL-LE.

Many ice sheet model projections included in ISMIP6 may be missing an important contributor to ongoing Greenland ice loss by not forcing a match to the initial transient of the Greenland Ice Sheet. If the ice mass change in CTRL-LE is subtracted

from WARM-LE similarly to ISMIP6, the resulting anomalous mass loss in WARM-LE in 2100 is ~70 mm SLE, which would fall just below the median of the range of model sensitivities in ISMIP6 (40-138 mm SLE). However, as we have explained above, this is strongly a function of our initialization scheme producing a strong transient ice loss even in the control setting,





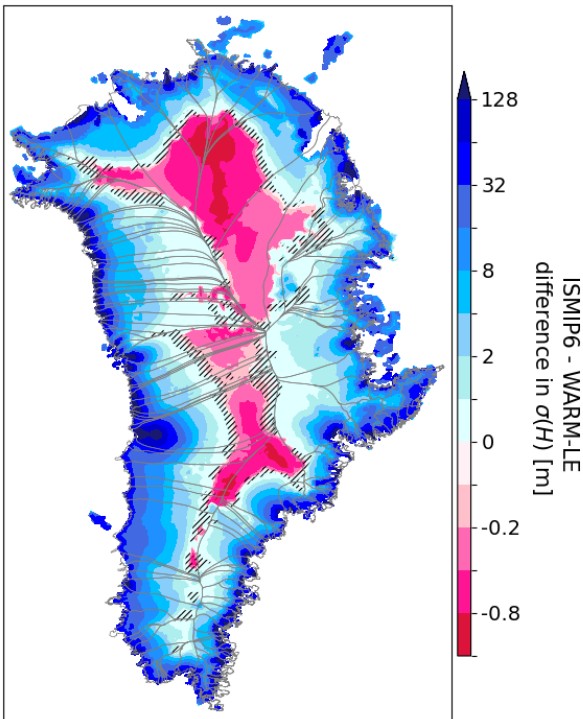

**Figure 14.** Difference between ice thickness spread ($\sigma(H)$) in 2100 for the ISMIP6 inter-model ensemble forced by MIROC5-RCP8.5 (Figure 6b in Goelzer et al., 2020) and ice thickness spread in 2100 for WARM-LE. Blue indicates greater spread for ISMIP6, pink for WARM-LE. Note the non-linear and non-symmetric colorbar. Hatching denotes non-significant differences, evaluated with a 300-sample bootstrap procedure for WARM-LE only, and controlling for a false discovery rate of 0.05 (see Appendix G).

without mean climate change. Without subtracting control simulations, ice mass loss in WARM-LE would fall near the upper end of the range of ISMIP6 model sensitivities with ∼120 mm SLE. The IPCC handled this lack of capability by subtracting

control simulation from ISMIP6 simulations under future forcing and then adding the current ice sheet mass balance trend. This potential missing piece of simulated ice sheet mass loss in ISMIP6 has previously been identified by Aschwanden et al. (2021), and is a focus of efforts to improve ensemble design in the coming ISMIP7 effort.

  Our initialization utilizes state-of-the-art methods (Choi et al., 2021) to match recent trends at both the ice sheet and glacier scales. This procedure relies on the calibration of a stress threshold parameter in the von Mises calving law (Morlighem et al.,

2016). However, since this is performed for each individual marine-terminating glacier, the number of degrees of freedom involved is as large as the number of glaciers, rendering our calibration prone to over-fitting. Furthermore, given that marine glaciers have response timescales on orders of decades to millennia (Robel et al., 2018), matching 10-year observed trends is no guarantee of realistic model behavior. For example, we urge caution in interpreting the dramatic retreats of PG, ZI, or SK as realistic projections; instead, we suggest that these may reflect limitations of current tuning practices in ice sheet modeling.



## 4.3 Implications

One of the most significant results in this study is that when glaciers with significant potential for contribution to sea level rise undergo rapid retreats, uncertainty in total ice sheet mass loss rapidly increases. This rapid increase in uncertainty is predicted by theory (Robel et al., 2019) and has occurred in other parameter-perturbed ice sheet model experiments (DeConto et al., 2021; Lowry et al., 2021). However, because the retreats simulated for large glacier catchments in this study are short-lived, this elevated ensemble spread is equally short-lived when all ensemble members ultimately go through retreat. This is because the retreats simulated in this study are the combined result of increased flow and calving rates on short (<10 km) portions of bed topography that deepen towards the ice sheet interior. In Antarctica, many ice sheet models predict prolonged periods of rapid retreat over long overdeepenings (100s of km; Seroussi et al., 2020, 2024). Thus, we may expect that a similar large ensemble study conducted for Antarctica could produce decades-centures with strongly elevated ensemble spread. Prior idealized large ensemble simulations of Thwaites Glacier, or representative configurations of it, show that its retreat under climate variability exhibits such an elevated ensemble spread over periods of decades to centuries (Robel et al., 2019; Hoffman et al., 2019; Bradley and Hewitt, 2024).

Our results indicate that the transient initial ice sheet tendency drives ensemble-mean behavior for multiple decades before the mean climate forcing begins to play a more prominent role in controlling ice sheet behavior. This result highlights the critical role of initializing ice sheet models to match recent observed ice sheet behavior, which should ultimately reduce inter-model spread by providing a common point of comparison at least at the outset of simulations. The role of initialization can be hard to discern in multi-model ensembles such as ISMIP6 (Goelzer et al., 2020) where models adopt very different initialization procedures, or in parameter-perturbed ensembles where parameter changes produce artificial disequilibria with initial model state (DeConto et al., 2021). This will have the greatest influence on uncertainty in ice sheet evolution over time scales of decades, which are the most important for communities planning for sea level rise from ice sheet melt (Bassis, 2021). However, the absolute values of such uncertainties are ultimately just 1-2 mm sea level equivalent which — like other sources of global sea level projection uncertainty — are relatively small compared to short time-scale uncertainties in coastal processes.

A final important finding of this study is that, at the scale of the Greenland Ice Sheet, SMB variability is the single largest component of climate variability contributing to ensemble spread. Under warming conditions, both increasing amplitude and spatial correlation of SMB variability also substantially contribute to ensemble spread. The non-stationarity in SMB variability has been observed in recent ice sheet mass balance (Boers and Rypdal, 2021) and surface mass balance (Slater et al., 2021), and is predicted to continue into the future by models (Fyke et al., 2014). It has been shown that such non-stationarity is expected as a result of increased poleward moisture transport (Bintanja et al., 2020).

An intercomparison of SMB models (Fettweis et al., 2020) show that dEBM generally compares as well to 30 years of Greenland SMB observations as most other high-fidelity SMB models. Additionally, at the few locations of available high-resolution ice cores, Ultee et al. (2024) found that high-fidelity SMB models appear to represent the time scales of variability in a reasonable fashion. However, the lack of high-quality long-running SMB observations means that it is not possible to statistically validate the amplitude or time scales of variability simulated in dEBM across the entire ice sheet (and by extension





AWI-ESM) in a robust fashion. Similarly, for *TF*, we rely on the time scales of variability simulated in AWI-ESM as there
are almost no locations around Greenland with continuous records of ocean temperature at depth for several decades. The
potential importance of time-varying SMB variability to uncertainty in ice sheet projections suggests that SMB models need to
be calibrated, not just to match the mean, but also to capture the amplitude and spatial correlation of interannual variability, and,
critically, their multi-decadal changes. Using novel techniques for measuring SMB changes from ice cores at high temporal
resolution (Trusel et al., 2018) will thus likely be critical to determining how SMB variability may change in warmer climates.

## 735 **5 Conclusions**

Internal climate variability has played an important role in mass change from the Greenland Ice Sheet in the industrial era. In
this study, we have shown that on decadal time scales, such aleatoric uncertainty will be an important contributor to projected
ice sheet mass loss, regardless of anthropogenic emissions. These time scales are important for planning in coastal regions.
Beyond a few decades, internal climate variability continues to drive increasing uncertainty in ice sheet simulations, but at a
level that is relatively limited compared to uncertainty from anthropogenic emissions and modeled ice sheet processes. During
rapid glacier retreats, aleatoric uncertainty can grow rapidly, but in our simulations, these periods are short-lived and the effect
on ice loss uncertainty is localized to a few catchments in West and Northern Greenland. Simulations of marine-terminating
glaciers in Antarctica often produce more long-lived retreats due to differences in ice sheet geometry, thus raising the possibility
of substantial and sustained aleatoric uncertainty in future ensemble projections of Antarctic ice loss.

While the representation of variability in climate models has improved substantially, in order to validate these models
and calibrate stochastic parameterizations properly, high-quality observations are necessary. This study provides additional
motivation for continuing and expanding such critical remote sensing and field work efforts for the purpose of improving
projections of future ice sheet change and uncertainty therein.

   Beyond quantifying the aleatoric uncertainty in simulations of future Greenland Ice Sheet evolution, another central aim of
this study is to provide a dataset for the research community to aid in answering questions around the "natural" envelope of
ice sheet and glacier variability in Greenland. The ensemble outputs are provided as open-access datasets, with information on
access and data analysis detailed in the code and data availability section. We hope that this resource will support continuing
community efforts on the critical task of quantifying and reducing uncertainty in projections of future ice sheet change.

*Code and data availability.*    All outputs from StISSM included in GrISLENS (450 GB in size) are archived as NetCDF files in an open-access
repository at the Arctic Data Center: https://doi.org/10.18739/A2VX0651F. All figures in this study can be reproduced with the GrISLENS
data repository and code included in the above-linked repository. Upon publication of this manuscript, an interactive tool on the CryoCloud
platform will be available to manipulate and plot GrISLENS output completely in the cloud.



## Appendix A: Calculation details

### A1 Thermal Forcing (*TF*)

We calculate *TF* with a salinity- and depth-dependent empirical equation for the freezing point (Cowton et al., 2015):

$$TF(\mathbf{x},t) = T_{oc}(\mathbf{x},t) - (\lambda_1 S_{oc}(\mathbf{x},t) + \lambda_2 + \lambda_3 z), \tag{A1}$$

where $T_{oc}$ is the ocean temperature [°C], $S_{oc}$ is the ocean salinity [psu], $z$ is the vertical coordinate with respect to the surface level [m, positive upwards], and $\lambda_1, \lambda_2, \lambda_3$ are parameters set to $-5.73 \times 10^{-2}$ °C psu$^{-1}$, $8.32 \times 10^{-2}$ °C, and $-7.61 \times 10^{-4}$ °C m$^{-1}$, respectively. The dependence on space and time is highlighted by **x** and *t*, respectively. Note that in Eq. (A1), *TF* is the

depth-specific value, but in the main text, we use *TF* to indicate the depth-integrated value.

### A2 Autoregressive Moving Average process (ARMA)

An ARMA process for a given variable $y$, of autoregressive order $p$ and moving-average order $q$, is denoted as ARMA($p$,$q$) and formulated as:

$$y_t = \sum_{i=1}^{p} \varphi_i y_{t-i} + \sum_{j=1}^{q} \theta_j \epsilon_{t-j} + \epsilon_t, \tag{A2}$$

where the $t$ subscript denotes the time step, and the AR and MA coefficients are denoted by $\varphi_1, ..., \varphi_p$, and $\theta_1, ..., \theta_q$, respectively. The $\epsilon_t$ term is a Gaussian noise term. The $\varphi_i$ coefficients capture the memory of the ARMA process, and the $\theta_i$ coefficients capture the persistence of random noise effects on the process evolution. We specify covariance between the different stochastic climate forcing variables through the Gaussian noise terms in the ARMA models. Specifically, a vector $\boldsymbol{\epsilon_t}$ can be drawn from a multivariate Gaussian distribution of dimension equal to the number of processes of interest:

$$\boldsymbol{\epsilon_t} \sim N(0, \Sigma), \tag{A3}$$

where $\Sigma$ is the covariance matrix.

Figure A1 shows the selected ARMA combinations for the residual variability time series in *TF*, SMB, and runoff, and for all the catchments. The Bayesian Information Criterion (Schwarz, 1978) is used for the selection procedure.

To generate stochastic annual time series of our three climatic variables (*TF*, SMB, and runoff), we use different covariance

matrices ($\Sigma$ in Eq. (A3)) for our different sub-periods separated by the 2000, 2050, and 2100 breakpoints. Given a correlation matrix $C$, a corresponding covariance matrix can be computed as:

$$\Sigma_i = K_i C K_i, \tag{A4}$$

where the $i$ subscript stands for the $i$th sub-period, and $K_i$ is a diagonal matrix with the marginal standard deviations of the $i$th sub-period on the diagonal. While the correlation matrix $C$ used to compute the different $\Sigma_i$ matrices remains identical,

the amplitude of the variances changes through changing $K_i$. Therefore, we need to scale appropriately the marginal standard



deviation of all $\epsilon_t$ terms (Eq. (A2)), denoted here $\sigma(\epsilon)$. In particular, $\sigma(\epsilon)$ is not equal to the standard deviation of the $y_t$ process (Eq. (A2)), denoted $\sigma(y)$. Any $\sigma(y)$ can simply be computed from a given time series, but the entries of $K_i$ must be $\sigma(\epsilon)$. Thus, we describe here our method for estimating $\sigma(\epsilon)$ for all sub-periods of all the 676 time series. For a given time series, we first calibrate the selected ARMA model (see Fig. A1) to the full time series. If both the $p$ and $q$ orders (Eq. (A2)) are at most 1, there is an analytical expression to calculate $\sigma(\epsilon)$ (e.g., Wilks, 2011):

$$\sigma(\epsilon) = \sigma(y)\sqrt{\frac{1 - \varphi_1^2}{1 + \theta_1^2 + 2\varphi_1\theta_1}}, \tag{A5}$$

where $\varphi_1 = 0$ if $p = 0$ and $\theta_1 = 0$ if $q = 0$. If $p$ or $q$ is greater than 1, we estimate $\sigma(\epsilon)$ numerically. We generate 10 time series from the calibrated ARMA process, starting with $\sigma(\epsilon) = \sigma(y)$. We iteratively decrease the estimate of $\sigma(\epsilon)$ until the mean total standard deviation of the 10 time series agrees with $\sigma(y)$ within 0.1 %.

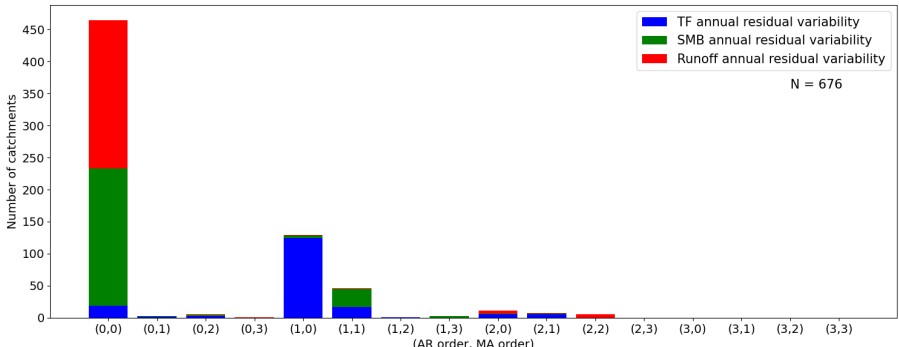

**Figure A1.** Histograms of best fitting ARMA models for the 676 z-score time series. Results for the *TF*, SMB, and runoff time series are shown in different colors. Selection is based on the BIC. The autoregressive order (AR) and the moving-average order (MA) correspond to $p$ and $q$ in Eq. (A2), respectively. Results for AR or MA orders of 4 are not shown, because they have histogram counts of 0.

## Appendix B: SMB-elevation profiles

Within-catchment spatial variability of SMB is captured through estimation of the slope of SMB-elevation profiles, referred to as lapse rates (e.g., Ultee et al., 2024). We compute lapse rates as the coefficients of a piecewise-linear regression of SMB versus elevation. We derive these functions using the dEBM SMB fields averaged over 2007 to 2203, because this corresponds to the period of our ice sheet simulations, as detailed in Sections 2.3 and 2.4. We use two elevation breakpoints in our piecewise-linear functions, thus resulting in three separate lapse rates, which capture the typical pattern of strong SMB decrease with elevation in the ablation zone, smooth increase in the lower-acummulation zone, and gradual decrease at the upper-most elevations. In Figure B1, we illustrate our fitting of lapse rate functions for 4 large GrIS catchments: Zachariae Isstrom, Helheim, Sermeq Kujalleq (Jakobshavn), and Humboldt. For catchments where the fitting procedure results in two





breakpoints separated by less than 100 m of elevation, we use a single breakpoint. Similar to the lapse rate values, the elevation
breakpoints are specific to each catchment.

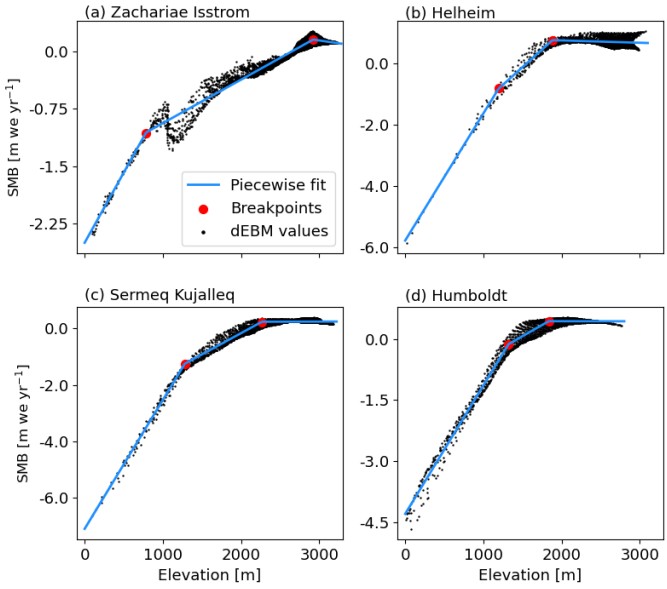

**Figure B1.** Illustration of the SMB fitting as piecewise linear functions of elevation. The dEBM mean 2007-2203 values are shown as black
dots, and the fitted lapse rates as the blue curves. The catchments shown are (a) Zachariae Isstrom in northeast Greenland, (b) Helheim in
southeast Greenland, (c) Sermeq Kujalleq in central-west Greenland, and (d) Humboldt in northeast Greenland.

### Appendix C: *TF* calculation caveat

In this study, *TF* is depth-integrated from the surface ($z = 0$ m) to the effective depth corresponding to the fjord sill ($z_{sill} < 0$
m) following: $TF = |z_{sill}|^{-1} \int_{z_{sill}}^{0} TF(z)dz$, where *TF* is the depth-integrated value, while $TF(z)$ is the depth-specific value. Our
initial objective was to follow the procedure of Slater et al. (2020). However, Slater et al. (2020) not only integrate until $z_{sill}$,
but also set the water properties below $z_{sill}$ down to the glacier front depth ($z_{front} < z_{sill}$) to the water properties at $z_{sill}$. They
then integrate *TF* over $z_{front}$, by computing: $TF = |z_{front}|^{-1} \left[ \int_{z_{sill}}^{0} TF(z)dz + TF(z_{sill}) (z_{sill} - z_{front}) \right]$. We realized our error in
not following the expression of Slater et al. (2020) only after the 372 simulations of this study had been performed, and we
acknowledge this caveat in our results. However, we also believe that the impact of this caveat in the interpretation of our
results is minor to minimal for the following reasons. (i) The assumption of within-fjord waters being linearly stratified and
perfectly homogeneous below $z_{sill}$ is subject to large uncertainties, due for example to air-sea heat exchanges, within-fjord
mixing processes, glacially modified water, and iceberg melting (e.g., Jackson and Straneo, 2016). The discrepancy between
our *TF* calculation and the one from Slater et al. (2020) is likely much smaller than the uncertainty range about the true *TF*.
(ii) Our computation of *TF* is consistent across all simulations, which implies that temporal variability and trends in *TF* are




consistent with those simulated by AWI-ESM. (iii) Given that deeper waters on the Greenland shelf are typically warmer than
surface waters, not integrating below $z_{sill}$ likely slightly underestimates *TF* compared to the method of Slater et al. (2020).
However, we calibrate (see Sect. 2.3.2) the calving sensitivity of all marine-terminating glaciers to their respective retreat rates
over the observational record. As such, if there is any underestimation of the true *TF* at a given glacier, it is likely compensated
by an overestimation of calving propensity; and note that frontal melting and calving at the terminus are treated identically by
our ice sheet model.

## Appendix D:  Empirical sparse correlation matrix


Figure D1 shows the sparse correlation matrix between climate forcing variables and glacier catchments discussed in the
Methods.

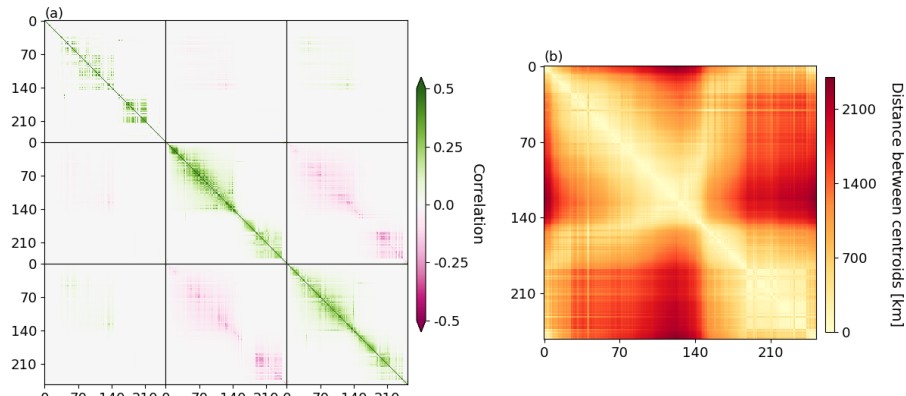

**Figure D1.** (a) Sparse correlation matrix of the residual variability in the three climatic variables (*TF*, SMB, runoff) for all 253 catchments.
Each black square along the diagonal represents correlation over the 253 catchments for a variable, while each off-diagonal black square
represents the cross-correlation over the 253 catchments for a pair of variables. The order of the climatic variables in the correlation matrix is:
*TF* (top and left), SMB (middle and middle), runoff (low and right). Black lines separate submatrices specific to a pair of climatic variables for
the 253 catchments. Catchment numbers correspond to a clock-wise arrangement of the catchment centroids, starting from the northernmost
catchment. Note that the colorbar saturates at 0.5. (b) Distance matrix between the centroids of the 253 catchments. The discontinuity around
catchment 140 corresponds to the shift between East- and West-Greenland.

## Appendix E:  Time evolution of standard deviation and skewness

Figure E1 shows the time evolution of the standard deviation and skewness for all large and small ensembles.



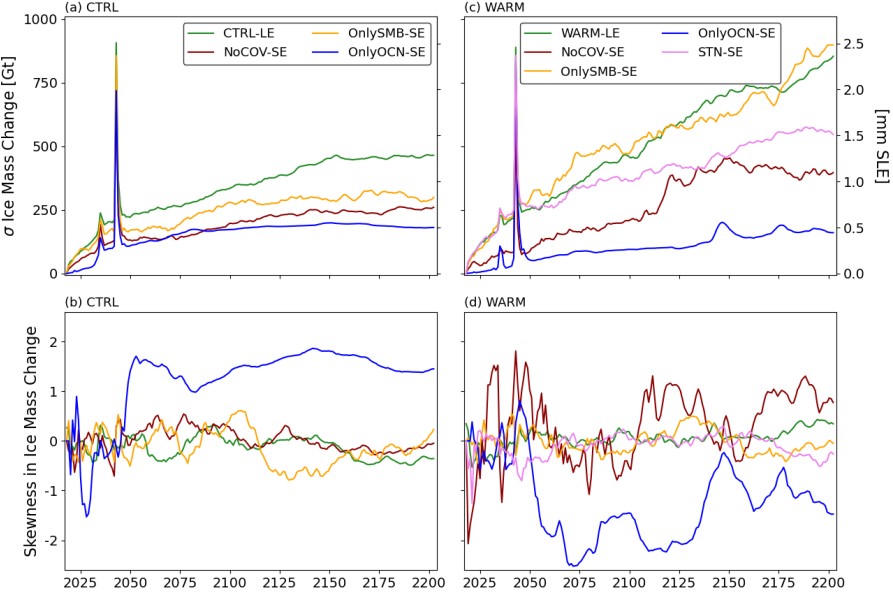

**Figure E1.** (a,c) Standard deviation and (b,d) skewness in ice mass change of the ensembles in the (a,b) CTRL climate forcing experiments, and (c,d) WARM climate forcing experiment. Note that y-axes are shared among panels of a same row. We recommend caution in interpreting the skewness of the small ensembles, which have only 30 members, because the skewness is a third moment statistic, and thus subject to large sampling uncertainty in small samples. See Sect. 2.4 for details about the large ensembles (LE) and small ensembles (SE). On the right y-axes, mm SLE denotes mm sea-level rise equivalent.

**Appendix F:  Greenland-wide climate forcing**

In this section, we illustrate the climate change forcing imposed under the RCP8.5 (WARM-LE) scenario compared to the pre-2000 (CTRL-LE) conditions. At the Greenland scale, the pre-2000 and post-2000 conditions can be compared in Fig. F1. The SMB time series is the annual Greenland-wide mean value, and the runoff time series is the annual Greenland total value. For the *TF* variable, we take the weighted mean of the individual marine-terminating glacier *TF* values, where weights are the area

of the catchment drained by each glacier. SMB and runoff values are from the dEBM output, and *TF* from the bias-corrected and extrapolated values of AWI-ESM (see Sect. 2.1).

**Appendix G:  Statistical significance testing**

When showing differences in ensemble mean or standard deviations between different ensembles, we test for statistical significance. We use a bootstrapping procedure with 300 bootstrap samples, each one consisting of a random sampling with

replacement of size equal to the original ensemble (100 for the large ensembles, 30 for the small ensembles). We compute the




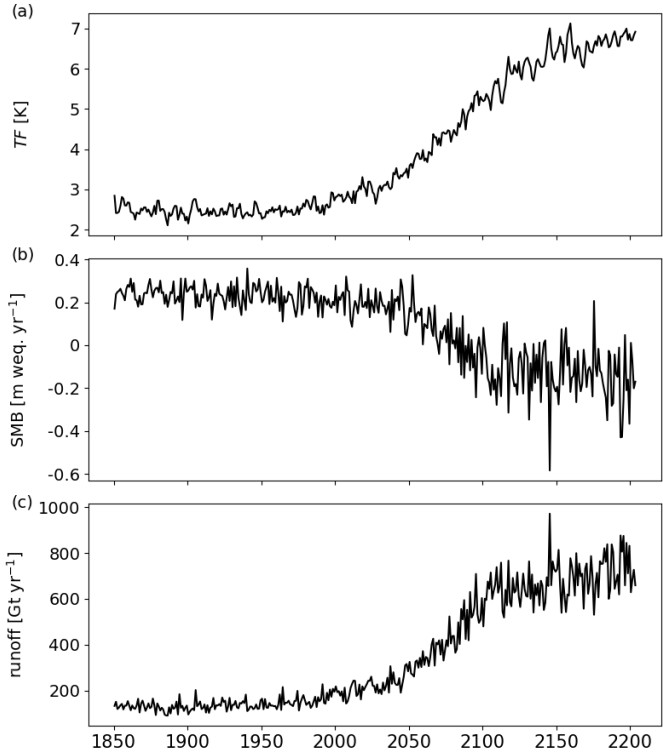

**Figure F1.** Greenland-wide values of (a) *TF*, (b) SMB, and (c) runoff. In (a), *TF* is the area-weighted average *TF* across all catchments with marine-terminating glaciers. In (b), SMB is the Greenland average. In (c), runoff is the Greenland total.

difference in the statistic of interest (mean and/or standard deviation) for each sample. We then compute the two-tailed $p$-value of the event zero difference.

For each Greenland map, there is one local statistical significance test per grid point. This situation is known as multiple hypothesis testing. We report statistical significance by controlling for a false discovery rate (FDR) of 5%. The FDR approach
adjusts for test multiplicity by placing a strict limit on the fraction of significant grid cell results that are spurious (Wilks, 2016). In this procedure, we derive the critical $p$-value for a test being considered significant ($p^*_{FDR}$) following:

$$p^*_{FDR} = \max_{i=1...N} \left( p_i : p_i \leq \frac{1}{N} \alpha_{FDR} \right) \tag{G1}$$

where $N$ is the number of local hypothesis tests, $\alpha_{FDR}$ is the FDR level chosen (5% here), and the $p_i$ are the local test $p$-values sorted in ascending order ($p_1 < p_2 < ... < p_N$). This method ensures that the selected FDR level is the upper limit for
the overall expected proportion of erroneously rejected local null hypotheses among the rejections. This expectation holds regardless of the unknown proportion of local tests having true null hypotheses. In contrast, reporting significance on a local test by local test basis only controls the probability of each individual true null hypothesis being erroneously rejected. As such, there is no overall control, and the proportion of erroneously rejected null hypotheses is an unknown function of the proportion



of null hypotheses that should be rejected. Since the rejected hypothesis tests are of interest, i.e., non-zero difference, it is
preferable to control the proportion of those rejections that are meaningful. We refer to Wilks (2016) for details and examples.

*Author contributions.* VV, AAR, LU, HS and AT conceived the presented work and methodology. VV conducted all simulations and analyses. LA and UKK provided output from AWI-ESM and dEBM and provided critical consultation on these climate models. YC developed and provided consultation on the model initialization and calibration methdology. VV and AAR wrote the manuscript together. All authors contributed to editing the manuscript.

*Competing interests.* The authors declare that they have no conflict of interest.

*Acknowledgements.* We acknowledge the computing resources that made this work possible provided by the Partnership for an Advanced Computing Environment (PACE) at Georgia Tech in Atlanta, GA with computing credits provided through startup from the University System of Georgia. We would like to thank research scientist Fang (Cherry) Liu for her assistance on challenges related to PACE and HPC. We thank Heiko Goelzer for sharing Greenland ISMIP6 results. This manuscript details work on the Stochastic Ice Sheet Project
(StISP), a grant funded by the Heising-Simons Foundation (no. 2020-1965). VV, AAR, LU, HS and AT were funded as a part of StISP. HS acknowledges support from the Novo Nordisk Foundation under the Challenge Programme 2023 - Grant number NNF23OC00807040. UKK is supported by the Deutsche Forschungsgemeinschaft (Excellence Cluster "EXC 2077: The Ocean Floor - Earth's Uncharted Interface", project no. 390741603 ) and acknowledges the Helmholtz Climate Initiative REKLIM (Regional Climate Change), a joint research project of the Helmholtz Association of German research centers. LA acknowledges funding from the German Federal Ministry for Education and
Research initiative PalMod (project PalModIII 1.1, BMBF grant no. 01LP2313A).




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
