# Peer review of "The Greenland Ice Sheet Large Ensemble (GrISLENS): Simulating the future of Greenland under climate variability"

_EGUsphere, 2024_

## Author Comment (AC1)

We thank the editor and both reviewers for their thoughtful suggestions of this study. Below find reviewer comments colored blue and our responses colored **black and bolded**.

**Reviewer 1**

**General comments**

In this paper, the authors develop the first large ensemble calibrated with observations that resolves individual outlet glaciers, using a variable mesh that achieves resolutions of less than 1 km at the margins. They employ the Stochastic Ice-Sheet and Sea-Level System Model (StISSM) to simulate the Greenland Ice Sheet (GrIS). This study quantifies the effect of internal climate variability on the evolution of the GrIS over 185 years under two climate scenarios: RCP8.5 (WARM) and pre-2000 (CTRL) conditions. The authors use the AWI-ESM model for atmospheric and oceanic forcing, the dEBM for downscaling atmospheric forcing to 5 km, and introduce a stochastic component to the surface mass balance (SMB), thermal ocean forcing (TF), and runoff, both individually and in combination with different correlations. The study provides interesting conclusions and certainly relevant to the purpose of TC and is a valuable contribution to understanding the effects of stochastic climate variability on the GrIS evolution. Additionally, its clear and precise writing is appreciated. Therefore, I recommend the publication of this paper in TC after minor revisions, which I outline below.

**Thanks to this reviewer for their suggestions. We have provided responses to their suggestions below.**

Regarding the calibration and the initialization, I would like to raise a few points for discussion:

1. The calibration is performed using a deterministic simulation, whereas most of the simulations in this work include a stochastic component in the forcing. Why was this choice made? A brief justification in the text would be greatly appreciated.

**We have added a justification of this choice to the Methods section, as follows:**

*"This calibration run does not explicitly include stochasticity in the climate forcing, but does include the climate forcing exactly as simulated by AWI-ESM for this time period, which includes internal variability. Thus, any drift induced by variability during this period is retained (Robel et al. 2024). Stochastic forcing during this calibration run would necessitate a separate calibration for every ensemble member, which would produce parametric differences between ensemble members that are orthogonal to the scientific goals of this study. This is perhaps a limitation of calibrating over a short time period, which is discussed in section 4."*

2. Figure 2 shows the mass change estimated by the IMBIE (Otosaka et al., 2023) and that simulated by the model. In L317, the authors state that "the total modeled 2007-2017 mass change agrees with the observational record within uncertainty ranges, even though the modeled mass loss rate is over- and under-estimated in the early and later years [..]". This overestimation in the model's mass change in some years (~2008–2011) appears to exceed the IMBIE uncertainty range by a significant margin (up to ~400 Gt in some instances).

As described by the authors, there is strong agreement between the model and observations in the later years of the simulation (i.e. in the total mass change for this period, which was the goal of the calibration), but the rate of mass loss in the model appears to be steeper than in the observations and this could affect the future evolution of the ice sheet. Indeed, as discussed in the Discussion section (L672), this calibration is responsible for the loss of mass in the early decades of the CTRL-LE simulations, which is the most abrupt change occurring in all the simulations (both in the WARM and in the CTRL scenarios).

Given these points, I believe it would be beneficial to provide additional justification for the choice of this calibration, clarify the origins of the differences with observations when this difference is maximal (both spatially and in terms of physical mechanisms or model uncertainties.), and explain why this particular calibration approach was adopted despite the noted discrepancies with Otosaka et al. (2023) and the effects in the CTRL-LE simulations in the following years. Additionally, in the discussion (L691–L694), it is mentioned that the results for glaciers with more abrupt retreat should be interpreted with caution. I believe this paragraph could be expanded further, providing a more critical reflection on the results themselves and elaborating more on their limitations.

**These differences between the calibrated ice sheet transient trend and observations are fundamentally the result of three different aspects: (1) starting from the BedMachine ice thickness field in 2007, (2) calibrating over a time period (11 years) that is short relative to the natural response time scale of flow at glacier termini (multiple decades and longer, Robel et al., 2018), and (3) imperfect representation of ice sheet dynamics in the model.**

**BedMachine is built to respect mass conservation, but integrates data from many sources that are not exactly co-located in time and may not include accurate information about the tendency in ice thickness. Consequently, upon starting the simulation, there are likely to be transient adjustments in the first few years that are a result of artifacts in the BedMachine product and not reflecting the true ice sheet thickness tendency. These transient adjustments are furthermore compounded by the model converging to an ice geometry in balance with the stress balance approximation (SSA), and parameterizations used (e.g., the von Mises calving law). Our aim here is simply to use currently available best practices to arrive at the initial (2018) ice sheet state for the ensemble simulations, but there are known shortcomings with this approach that we take into account when interpreting our results. We've expanded on this further in the Methods section:**

***"This departure from the IMBIE observations of ice sheet mass loss is an unavoidable consequence of having a relatively short calibration period (11 years) in comparison to the natural response time of glacier termini (multi-decadal and longer time scales; Robel et al., 2018). Inaccuracies in the 2007 ice sheet state, which is derived from a mass-conserving data assimilation scheme (Morlighem et al., 2017), transiently adjust to the stress balance approximation, and to the assumption of the von Mises calving parameterization with calibrated values of $\sigma\_max$ at all glaciers. These imperfect initial conditions and model physics combine to cause the temporary departure from the observed mass loss trajectory."***

**In the discussion section, we have expanded on our cautionary note as follows:**

*"As discussed in section 2.4, the 2007 ice sheet state is derived from a mass-conserving data assimilation scheme that integrates data that is not exactly co-located in time, producing inaccuracies which likely transiently adjust during the calibration window and potentially beyond 2018 into the ensemble simulations. Additionally, the exact timing of simulated rapid retreats at PG, ZI, and SK depends on the calibrated values of parameters used in the von Mises calving parameterization. We recognize that all calving parameterizations currently used in models have known issues when extrapolated across space and time (Amaral et al., 2020). We thus urge caution in interpreting the timing and extend of dramatic retreats of PG, ZI, or SK as realistic projections; instead, we suggest that these may reflect limitations of current tuning practices in ice sheet modeling. However, we emphasize that the aim of this study is not to project exact ice sheet evolution into the future, but rather to quantify how uncertainty in ice sheet mass loss from climate variability is modulated by such rapid retreats and other processes."*

3. While the glacier retreat shown in Figure 3 is illustrative of the present-day performance, it would be valuable to also assess the model's pre-2000 performance against observations using a 2D plot of ice thickness and surface velocities, perhaps as an appendix. I am surprised that the western margin experiences almost no ice mass loss until 2050 in any of the simulations, considering that this region is currently one of the areas experiencing the highest ice loss (Mouginot et al., 2019). However, in the Calibration section, the authors note that they are unable to accurately represent the retreat of the SK glacier (Jakobshavn), and in the Discussion section, they further mention that the results for this glacier should be interpreted with caution. It would be helpful if the authors could elaborate on why they believe the ice sheet retreat in this region is not being adequately simulated.

**This is a good suggestion. We have added a figure in the appendix (Figure G1), reproduced here, showing thickness change during the calibration period (2007-2017) compared against observed thinning for this period from Khan et al. (2024). We find good agreement between the simulation and observations, particularly in reproducing thinning on the Western margins, including within the SK catchment. If anything, thickening in the interior is a bit greater than in observations, but this is also visually exacerbated by the log color scale we use here. The reviewer's perception "that the western margin experiences almost no ice mass loss until 2050 in any of the simulations" is not quite accurate. This perception may be influenced by Figures 7a and 8a, where the Western thinning rate is less compared to the rapid thinning (100s of meters) due to retreat of PG and ZI , which causes a wide color bar range. In the first 30 years of all our simulations simulation, ice in W GrIS thins by ~100 meters, which continues current rates of thinning in this region (1-5 m/yr), and probably just a continuation of the tendency captured in the ice sheet initial condition that we calibrate. Our desire to caution readers about SK is mainly that its modeled retreat is quite sensitive to choice of $\sigma_{max}$ in that catchment, and so further work is needed to improve calving parameterizations and calibrate them appropriately.**

[Figure]

Many results are presented, with extensive discussion on the spread within different ensembles, the factors that increase dispersion, when internal climate variability is more relevant, and the types of stochastic forcing that have the greatest impact on ice sheet development. These results are important and provide clarity on the problem and quantify the uncertainties associated with internal climate variability. However, one of the main scientific questions of the paper is the actual effect of adding stochastic forcing on the evolution of the ice sheet. For this reason, a more thorough comparison between the ensembles and the deterministic simulations is missing.

**We appreciate this suggestion and have made changes in response to the specific points below**

Therefore, it would be interesting to assign greater emphasis to the deterministic simulations shown at the beginning of the Results section and to compare the large ensembles with them in greater detail. In this way we could better see the differences between using a purely deterministic forcing and adding a stochastic component. Some recommendations for this include (which could also be incorporated into a separate figure or presented differently):

- Include these simulations in Table 1 where the experiments are summarized, using distinct names (e.g., CTRL-Det and WARM-Det).
  **Added**

- Include their profiles in Figure 6.

**We have included the profiles of the deterministic simulations as grey profiles in Figure 6.**

- In Figures 7 and 8, include the mass change (or the difference relative to the ensemble mean) for CTRL-Det and WARM-Det in the years 2050, 2100, and 2203, and discuss any spatial differences between the deterministic and stochastic simulations.

  **We have included these maps of thickness change of the deterministic simulations. As suggested by the reviewer, we prefer to show the difference between the ensemble mean and the deterministic simulation because these differences are much smaller than the temporal changes in ice thickness (1 to 2 orders of magnitude). As such, no difference between the upper and middle rows can be perceived if we simply show the deterministic ice thickness change in 2050, 2100, and 2203. We have added a brief discussion of those differences between CTRL-LE and CTRL-Det and between WARM-LE and WARM-Det.**

- Include in the figure 10 the value of the total ice mass change of CTRL-Det and WARM-Det.
  **We have added the CTRL-Det and WARM-Det mass change as a grey dashed line. Note also that we now show in Figure E1 a,d the anomaly of the ensemble mean of each small ensemble with respect to their corresponding deterministic simulation (CTRL-Det or WARM-Det).**

**Specific comments**
I will now provide a series of more specific comments on certain parts of the article that I believe could be improved.

L20: When I read the abstract, I did not fully understand the mention of the Antarctic Ice Sheet until I read the entire article. Therefore, I recommend either removing it from the abstract or adding a sentence explaining why studying internal variability in Antarctica is important, as done in the Implications section.
**We've modified the last sentence of the abstract to simply indicate that we argue for "extension" of these methods to the Antarctic Ice Sheet. This way, it flows more naturally in the context of the abstract's focus on Greenland.**

Regarding the CTRL simulations, I understand from the description in the Methods section that the forcing applied in these simulations is the same as in the calibration for the years 2007–2017, and that afterward, the mean forcing from 1850–1999 is used. Does this mean that an instantaneous cooling occurs in 2018 (with a drop in TF and runoff and an increase in SMB), returning to pre-2000 values? It would be helpful if authors included the time series of TF, SMB, and runoff applied to the CTRL simulations in Figure F1.
**Reviewer 1 is correct. The CTRL simulation use the statistics of the 1850-1999 conditions, and this starts immediately after the calibration (2007-2017). This abrupt transition is however of small magnitude compared to inter-annual variability for SMB, and of comparable magnitude to inter-annual variability for TF and runoff. We have updated**

**Figure F1, showing explicitly the 1850-1999 period, the calibration (2007-2017) period, and the transient (2018-2203) period with the mean 1850-1999 forcing shown for reference. Also, we now explicitly mention this aspect in the main text:**

*"Note that the pre-2000 climate conditions are applied immediately at the end of the calibration run, which induces an abrupt but low-magnitude change towards a cooler climate at the start of the transient simulations (Fig. F1)."*

**Finally, we note that any other method for transitioning from the calibration climatology to that of the CTRL period would have required assumptions on the artificial period for this transition. Here, we favored the most straightforward approach of using the CTRL period statistics immediately after calibration, since the magnitude of the climatic differences is small.**

Figure 4c: It is not entirely clear whether the peak around the year 2040 in both curves (WARM and CTRL) coincides exactly. It would be helpful to clarify this in the text or slightly adjust the way it is plotted.
**A sentence has been added in section 3.2 clarifying:** *"The two spikes in ensemble spread associated with these retreats occur at the same model years in CTRL-LE and WARM-LE."*

Figure 5: This figure is somewhat unclear, making it difficult to extract information from it. Additionally, compared to Figures 7 and 8, the only extra information it provides is a higher temporal resolution of the results glacier by glacier. Therefore, I would recommend modifying it for greater clarity. Below are some suggestions:
- I am not entirely sure what is shown in the color bars. It represents the terminus retreat, but some glaciers (Figures 5a and 5b) appear in red and then turn gray, which, according to my understanding of the figure caption, would mean they initially retreat and later experience no further advance or retreat. Meanwhile, other glaciers (such as those in the northwest) remain dark red throughout the entire time series until the year 2203. What does it mean when they stay dark red for the entire period? Additionally, it seems that many glaciers retreat by around 50–100 km before 2050, which is a surprisingly large value.

  **Yes, the figure refers to the cumulative retreat at any given year since 2018. We have clarified this in the caption:**

  *"Cumulative retreat of terminus position since transient simulation start date (2018) for each marine-terminating glacier in Greenland: a retreat of x km at year y indicates that the terminus has retreated by x km between 2018 and y."*

  **The grey hatching indicates that less than 95% of the ensemble members agree on the sign of retreat. In a 100 member ensemble, this metric can be sensitive to the slight re-advance of a small number of ensemble members. For example, in the case where most ensemble members are stable at their front position, but a few ensemble members are retreated to the next upstream bed peak, the ensemble mean would be red (driven by those retreated members). As the stable members oscillate about their front position, when sufficient members are slightly retreated about this**

**position, there is no grey hatching, and when too many members are slightly advanced about this position, there is grey hatching. We note however that such transitions from significant to non-significant retreat occur only for few glaciers, and only in cases where the ensemble mean retreat is of small magnitude.**

**Finally, we thank Reviewer 1 for pointing out the unrealistic retreat rates shown in the previous version of the figure. We realized that there was an error in our algorithm for diagnosing retreat rates from the output files, which affected many glaciers. Note that this error only affected the code specific to making Figure 5. The figure has been corrected accordingly, and retreat values are now smaller and more realistic. By 2050, only those glaciers which are affected by high sensitivity to the tuning in the calibration period (Petermann and Zachariae Isstrom most notably) have retreat rates exceeding 50 km by 2050.**

- In L485–L486, it is mentioned that the glaciers in the figure are numbered, but I am unsure what this refers to or whether the numbering is located elsewhere. In any case, it would be helpful to include a clear indication in the figure referencing these glaciers. Additionally, it would be useful to label the most relevant glaciers in the same way they are highlighted in Figure 7.
  **This numbering was a typo that remained from a previous version of a draft manuscript. It has been removed from the main text, and we instead refer to the corresponding Greenland regions. Note also that the locations of the glaciers discussed in the main text are explicitly shown in the map that has been added to Figure 5.**

- The GrIS is divided into the regions indicated in the left margin; including a small map showing this division would help better locate the glaciers and improve the overall understanding of the figure.
  **We have added a Greenland map in Figure 5, which shows the regions and glaciers discussed.**

- Finally, to maintain consistency with Table 1, I believe the figure titles should be labeled as CTRL-LE and WARM-LE instead of CTRL and WARM.
  **We have changed the sub-figure labels to CTRL-LE and WARM-LE.**

Regarding the results of the small ensembles shown in Figure 10, it would also be valuable to include the time series of ice mass change from these experiments for better visualization, either in the same figure or in E1.
**We have included the time series of the small ensemble ice mass changes in Figure E1 (a,d). We show the anomaly of the ensemble means with respect to the corresponding deterministic simulation (CTRL-Det and WARM-Det). Using the anomalies allow us to visualize differences between the small ensembles, because the absolute ice mass changes from 2018 to 2203 are orders of magnitude larger than the inter-ensemble differences. Thus, the y-scale required to show absolute changes would not allow to visualize any inter-ensemble differences. Note also that we prefer to show the ensemble mean to avoid showing**

**too many individual lines with different colors in single sub-figures, and because the standard deviation shown in (b,e) quantifies spread within single sub-ensembles.**

In L615–L617, the effects of oceanic variability are discussed. Although they are generally smaller than those of atmospheric variability, they prevail in the western region. It would be interesting to comment on why oceanic variability has a greater influence in these areas. This is expected, as shown in studies such as Slater and Straneo (2022), since although atmospheric forcing currently dominates GrIS mass loss, ocean warming has a greater influence on glaciers in the west and south.

**This was a great suggestion to cite Slater and Straneo (2022). We have also added a reference to Verjans et al. (2023), where we previously showed that seasonal and interannual ocean variability in these regions are highest in West and South Greenland. We've added a discussion to this effect here:**

***"This future sensitivity of glaciers in West Greenland to ocean forcing is consistent with a similar sensitivity of recent (1979-2018) glacier retreat to ocean variability in Central West and South Greenland, as shown by Slater and Straneo (2022). They suggest that South and Central West Greenland are directly exposed to North Atlantic oceanic variability. Indeed, the amplitude of both seasonal and interannual variability in our stochastic parameterization of the oceanic thermal forcing, derived based on a high-resolution ocean reanalysis product (Nguyen et al., 2012), are the highest in South and West Greenland (Verjans et al., 2023)."***

**Technical corrections**
L755 The creation of an open large ensemble dataset for the community is very much welcomed and useful; however, the link (https://doi.org/10.18739/A2VX0651F) to the repository does not work (last checked on February 22), and this should be fixed before the paper is published.
**Our apologies. Due to the size of the dataset (500 GB!) we had to spend extra time working with the team at the Arctic Data Center to get the dataset hosted properly. This link is now working.**

In some figures I have the impression that the limits in the colorbar have not been well applied and it appears cut in the Petermann and Zachariae Isstrom glaciers (figures 7d, 7e, 7f, 8d, 8e, 8f, 9, 11, 14).
**Yes, this is correct. We have deliberately chosen to saturate the colorbars. This is indicated by the upward and downward pointing arrows on the colorbars. This choice is motivated by the very large thickness changes at Petermann and Zachariae Isstrom glaciers, and/or other localized statistical outliers. If the colorbars show the full extent of values, most of the maps are simply uninformative because colors cannot be distinguished. As such, in the interest of clarity and legibility, we saturate the colorbars.**

L210: There is a mistake in the sentence "correlation patterns are strong within East- and West-Greenland, but weaker between East- and West-Greenland". Where exactly are the patterns strong?
**We can see how the sentence was confusing. We have re-phrased: *"For all variables, correlation patterns between glaciers close to each other are strong, however, correlations are weak over wider spatial scales, particularly between East and West Greenland."***

L319: "Figure 3 compares the observed and modeled 2007-2017 retreat rates". However, the figure does not show retreat rates, it shows the total retreat in kilometers.
**Fixed**

L177 L196 L316 L319 L374 L405 L413 L418 L426 L430 L482 L492 L526 L563 L777 L802 Instead of "Figure" it should be "Fig.".
**Fixed**

**References**
Mouginot, E. Rignot, A.A. Bjørk, M. van den Broeke, R. Millan, M. Morlighem, B. Noël, B. Scheuchl, & M. Wood, Forty-six years of Greenland Ice Sheet mass balance from 1972 to 2018, Proc. Natl. Acad. Sci. U.S.A. 116 (19) 9239-9244, https://doi.org/10.1073/pnas.1904242116 (2019).
Slater, D.A., Straneo, F. Submarine melting of glaciers in Greenland amplified by atmospheric warming. Nat. Geosci. 15, 794–799 (2022). https://doi.org/10.1038/s41561-022-01035-9

---

## Author Comment (AC2)

We thank the editor and both reviewers for their thoughtful suggestions of this study. Below find reviewer comments colored blue and our responses colored **black and bolded**.

**Reviewer 2**

In `The Greenland Ice Sheet Large Ensemble: Simulating the future of Greenland under climate variability,' Verjans and co-authors use a stochastic variant of the Ice Sheet System Model to explore the sensitivity of the Greenland Ice Sheet to variability in oceanic and surface mass balance forcing. In particular, they aim to quantify the relative importance of such so-called `aleatoric' uncertainty relative to other types of uncertainty derived from imperfect or unresolved modeling assumptions and initial conditions. Through a detailed comparison of ensemble experiments meant to represent both a continuation of contemporary forcing alongside a potential high end warming scenario, they find that the influence of stochastic climate is non-negligible over the coming two or so decades (in terms of total predicted mass change), while these stochastic effects become relatively unimportant over century-scales.

This result is interesting (albeit not particularly surprising) in that it illuminates a principal challenge for short term sea-level prediction, while providing some important guidance as to whether short time-scale variability represents a source of uncertainty that needs to be better quantified for long term projection (thankfully not, it seems!).

This manuscript represents an impressive and insightful culmination of several methodological threads that seem to have been `in the works' for a few years -- the development of StISSM and its ensemble generation tools, the statistical characterization of climate variability in a generative sense, and the coupling of ice dynamics to downscaled surface mass balance and frontal ablation paramterizations. The current work is undoubtedly at the vanguard of ensemble methods for ice sheet uncertainty quantification, and a big step forward for understanding Greenland's sensitivity to climate noise. I have no issues with the paper's general methodology. I have included below a few comments that I hope can improve the manuscript's clarity and utility.
**We thank the reviewer for their suggestions and address them each in turn below.**

L42: `are performed' should be `have been performed' for consistent case.
**Fixed**

L46: Perhaps here, perhaps elsewhere, it's maybe worth providing a higher level overview of where climate stochasticity comes from (and where it does not). In particular, it's worth noting that climate is very likely not actually random, but rather appears that way due to the chaotic dynamics characteristic of the atmosphere (and ocean, to a lesser extent). Ice sheets do not exhibit such ostensible stochasticity (EISMINT2 and ice streams notwithstanding), so the irreducible uncertainty in the ice sheet context is derived solely from the forcing term.

**These are good points, we have added additional sentences here to make the origin of this idea, that stochastic climate forcing can be used to approximate the deterministic chaotic behavior of climate variability, more clear. Here is the revised text:**

*"For any given scenario of anthropogenic forcing, there remains an ``irreducible'' (also called ``aleatoric'') uncertainty associated with the ice sheet response to internal climate variability, due to the limited predictability of the chaotic climate system (Lorenz 1969). Indeed, nearly 50 years ago, in work that would later garner a Nobel Prize, Hasselman (1976) posited that such chaotic deterministic climate variability could be equivalently represented by stochastic random forcing when simulating slow ``integrators'' in the Earth system, such as ice sheets. However, in the last five decades, there have been no sustained attempts to build such a stochastic large-scale ice sheet model."*

L84: What question is being referred to here?
**Changed to** *"We apply a novel stochastic modeling approach to produce this ensemble."*

L143: A qualitative description of what EN4 is, and why it's helpful for bias correcting the ocean thermal forcing would be very helpful.
**We have added additional details about the EN4 product here and why it is helpful for the bias correction used here (and refer again to Verjans et al. 2023 for more extensive discussion of this point).**

L202: I was expecting a similar interpretation of the moving average component of the fits. Do these exhibit any interesting patterns? Does the MA component even matter?
**We've added a sentence addressing this issue:** *"The fact that very few of the best-fitting parameterization include a moving-average component (q > 0) is not surprising and follows considerable prior work (Gilman et al., 1963; Hasselmann, 1976) showing theoretically and empirically that climatic variables are well described as purely stochastic autoregressive processes due to the memory implicit in systems with finite heat capacity."*

L203: I spend more time than most glaciologists thinking about covariance, and yet I'm still unclear as to what's going on here. In particular, after fitting the ARMA model to each time series of TF, SMB, and runoff independently, how are spatio-temporal correlations between them calculated. Reading the appendix, it seems that there are three layers to this model: Fitting a piecewise linear function, fitting an ARMA model to each basin/variable, and then computing a big covariance matrix between the residuals for all? Okay, I guess, but I would like a more centralized and coherent justification for why this is a reasonable way to control the spatial relationships.
**Yes, the reviewer is correct in their understanding of our spatio-temporal statistical modeling. The spatial unit level is the glaciological catchment. We have 253 catchments. For each of these, we have a time series of SMB, of runoff (for those catchments where runoff is non-zero), and of TF (for those catchments with a marine outlet). These time series are computed from the climate model outputs. Residuals from the variable- and catchment-specific ARMA models are correlated to compute the covariance matrix. Our covariance matrix is of dimension 676×676. We acknowledge that such a large covariance matrix is subject so spurious correlations when constrained using time series of only 354 yr (1850-2203). This limitation is discussed in the main text:**

*"The residuals are obtained after fitting the optimal ARMA model to a given time series, such that the stochastic component $\varepsilon_t$ (see Eq. A2) is isolated. Isolating residuals allows to first*

*remove potential spurious correlations between catchments and variables that appear in the raw time series caused by temporal autocorrelation. This autocorrelation is removed from the ARMA residuals, and the remaining correlations found capture cross-spatial and cross-variable dependencies. However, because the number of entries in the correlation matrix to be estimated (1/2 × 676 × 675 = 228 150) is large compared to the number of yearly samples (354), we compute a sparse correlation matrix (Hu and Castruccio, 2021) with the commonly-used graphical lasso method (Friedman et al., 2008)."*

**The sparse covariance matrix is shown in Figure D1. We understand the concern of validity of this approach. However, we justify our choice of using the glaciological catchment as a compromise between (1) using a large enough spatial unit, and (2) maintaining spatial detail at a level that is physically meaningful for glacier flow. Indeed, using the individual grid-point as a spatial unit would not only drastically increase the computational complexity of covariance estimation, but would also lead to increased risk of generating spurious correlations. By averaging over glacier catchments, we increase the signal-to-noise ratio of correlations between spatial units. Averaging at a coarser level would be possible, but at the expense of losing some of the details in SMB, runoff, and TF that partly explain different behaviors of glaciers located within a same region.**

**The motivation for fitting the covariance matrix to the ARMA residuals is the following: we need to separate temporal autocorrelation (within each catchment and variable) from the spatial cross-correlation (between different catchments and variables). Raw time series often exhibit strong temporal dependencies, which can artificially inflate estimates of spatial correlation if not accounted for. By modeling and removing the temporal structure via ARMA fits, the residuals represent the remaining component of each time series after accounting for its own past behavior. Computing covariance on these residuals ensures that the estimated spatial covariance matrix captures true cross-spatial and cross-variable dependencies, after having removed the temporal autocorrelation. It also enhances the interpretability and stability of the covariance matrix used downstream in the model. We have added this explanation in the manuscript, as given in the text snippet above (*Isolating residuals (…) cross-variable dependencies*).**

**This approach follows the framework laid out in Hu and Castruccio (2021), where controlling for temporal structure improves estimation of spatial dependencies in spatio-temporal datasets. It has subsequently been applied to Greenland SMB by Ultee et al. (2024) and Greenland TF by Verjans et al. (2023).**

Sec 2.2.3: I'm not completely sure that this is the right thing to do, but it might be helpful to lead the section with this (which is essentially the `physics'), so that the reader will have a better idea of what the TF, etc. is going to be used for. Similarly, you might include here the way that lapse-rates and such enter the SMB calculation.

**This subsection has been moved to the beginning of section 2 and a description of the SMB parameterization has been added.**

L264: I'm sympathetic to the need to use SSA for computational reasons, but it would be worthwhile to briefly describe the implications -- Greenland has a lot of ice that is very much not consistent with the assumptions of that model after all.

**Two sentences have been added addressing this assumption:** *"Using this approximation over the entire ice sheet may neglect deformation ice sheet flow, particularly in the ice sheet interior where ice is more likely to be frozen to the bed. However, over the centennial time scales considered in this study, these errors are unlikely to be significant on the scale of the entire ice sheet where most ice transport near the margins occurs via basal sliding."*

Eq. 3 and lines after: Am I missing a previous point at which $N$ is defined?  How is it computed here?  Constant fraction of overburden?

**We have added a definition and explanation of N, effective pressure, here.**

L271--273: Would it be possible to provide some additional justification with respect to the linear regression step described here?  This isn't something I've seen before, so it would be nice to understand a little bit better how/whether this works.

**Added citations and a justification for this approach:** *"This is a common approach in ice sheet model simulations (Åkesson et al., 2018; Cuzzone et al., 2022) where advance onto currently ice-free portions of the bed may occur. It captures the general pattern that deep portions of the bed are likely to have accumulated deformable marine sediments when they were covered by ocean rather than ice."*

Sec. 2.3.2: I am confused as to the technical approach for performing this calibration.  Is this done by manually fiddling with $\sigma_{max}$ until the eyeball norm is minimized, or is there an objective (and automated) procedure that is taking place?

**For calibration, we performed a large sequence of calibration runs. Between every run, we evaluated the retreat rates of all glaciers, and increased/decreased $\sigma_{max}$ of individual glaciers if their retreat had a positive/negative bias. This was performed until all glaciers reached a retreat rate within ±1 km of the observed retreat rate. For a subset of glaciers, this observational constraint could not be met while keeping $\sigma_{max}$ to physically acceptable values (see Figure 3). Once we achieved this objective, we compared the total ice sheet mass loss to the IMBIE mass loss. We then increased/decreased $\sigma_{max}$ at those glaciers where it was possible to still remain within the ±1 km individual glacier constraint in order to better match the IMBIE mass loss. The process was semi-automated: evaluation of retreat rates, of ice sheet mass loss, and adjustment of $\sigma_{max}$ values were automated. However, calibration runs had to be manually re-configured and re-launched. That process was quite tedious and work-intensive.**

L399: I am surprised that the assertion that the small deviation of the ensemble mean from the deterministic run is a result of noise-induced drift is not backed up by a statistical test.  It would strengthen the argument to include a test of significance here.

**You're right that it is not significantly different, which is what was implied by the following sentence. We have re-arranged and modified these sentences to be clearer that this difference does not represent a statistically significant difference. This is because the deterministic simulation can be considered as a single "sample" which falls well within the distribution of the stochastic ensemble. To be clear about our thinking: a statistical**

significance test (e.g., t-test) involves creating a statistic (e.g., t-statistic, if we are interested in the mean of a sample) that can then be compared to the expected distribution of such a statistic (e.g., t-distribution). In this case, we already have a sample (ice mass change in the deterministic run) and a distribution (ice mass change sampled by the stochastic ensemble members), which can be compared directly in a statistical inference problem, as we do in the current, now clarified, text. Thus, it should not be necessary to pursue more complicated statistical testing than this.

Here is the revised text:
*"By 2203, the mean ensemble ice loss is 4% greater than in the deterministic simulation. This difference is just 1 standard deviation of the ensemble final mass change from the ensemble median, and 17 members have less ice loss than the deterministic run. Thus, while this difference with the deterministic run may be suggestive of potential noise-induced drift in the stochastic ensemble (Tsai et al., 2017; Hoffman et al., 2019; Robel et al., 2024), it is not large enough to be statistically significant (i.e., differences of this magnitude or larger occur by chance in 17% of ensemble members). We disentangle the mechanisms of this drift more fully in Section 3.3 with comparison to small ensemble experiments."*

Fig. 4d: This is a challenging metric to use in order to assert the relative importance of uncertainty because the denominator gets so very small close to the start of the simulation period. I am not sure what the alternative is, but it might be helpful to acknowledge that.
**Yes, this is true. We have added a sentence acknowledging this:** *"We note that the denominator in this ratio starts near zero and so should be interpreted with caution, though it does help us to understand the relative importance of ensemble spread as compared to ensemble mean change."*

L465: delete `briefly'.
**Deleted**

L544: This is a pretty awkward sentence -- suggest rephrasing.
**Split into two sentences are reworded**

Discussion: I appreciate the comparison to both Tsai (for the forcing uncertainty comparison) and ISMIP6 (for the model uncertainty comparison), but it might be useful to also compare to some of the previous works that explore parametric uncertainty -- which seems to be of similar size to model uncertainty in some cases. Would using randomly sampled climate-to-SMB parameters drown out the influence of the stochastic climate? This would be important to know in making a decision about whether to include stochastic forcing in, say, ISMIP7.
**This is a nice suggestion. Aschwanden et al. (2019) is probably the most comparable single-model, parameter perturbed ensemble study. We have added a paragraph in the discussion drawing out this comparison, particularly with respect to what it means for designing ensembles to quantify uncertainty in future Greenland ice loss:**

*"Aschwanden et al. (2019) provides another useful point of comparison to GrISLENS. In that study, parameters for ice flow and forcing parameterizations were perturbed over 500 ensemble members with an ice sheet model of sufficiently high-resolution (1 km) over*

*Greenland to resolve individual outlet glaciers. They found consistently much greater mass loss compared to WARM-LE, with median sea level contribution of 22 cm in 2100 (compared to 11 cm in WARM-LE) and 103 cm in 2200 (compared to 25 cm in WARM-LE). The ensemble spread in Aschwanden et al. (2019) is also much greater than in WARM-LE, with a standard deviation of 10 cm SLE in 2100 and 50 cm SLE in 2200, about 2 orders of magnitude greater than the WARM-LE ensemble spread. Later work to calibrate these ensembles using observations reduced the median and spread of their ensemble (Aschwanden and Brinkerhoff, 2022), but the broader higher sensitivity and spread compared to WARM-LE remain. Decomposing the parametric uncertainty quantified in their ensemble using Sobol indices, they find that at 2100, uncertainty ice flow and surface melt parameters contribute the most to uncertainty in total ice sheet mass loss. At 2200 and beyond, uncertainty in the sensitivity of mean atmospheric temperatures to emissions forcing (i.e., climate sensitivity) dominates uncertainty in total ice sheet mass loss. Consistent with our study, they find that uncertainty in ocean forcing plays a relatively minor role in driving uncertainty in total ice sheet mass loss. We thus conclude that even for a given large-scale climate forcing, uncertainty in parameters that govern how SMB is calculated from large-scale climate models is currently large enough to substantially exceed uncertainty from variability in SMB in terms of the resulting influence on total ice sheet mass loss."*

---

## Author Response (AR2)

We thank the editor and both reviewers for their thoughtful suggestions of this study. We have made all the technical corrections suggested by the editor.